# Substrate stiffness governs the initiation of B cell activation by the concerted signaling of PKCβ and focal adhesion kinase

Samina Shaheen[1†], Zhengpeng Wan[1†], Zongyu Li[1†], Alicia Chau[2], Xinxin Li[1], Shaosen Zhang[1], Yang Liu[1], Junyang Yi[1], Yingyue Zeng[1], Jing Wang[1], Xiangjun Chen[1], Liling Xu[1], Wei Chen[3], Fei Wang[4], Yun Lu[5], Wenjie Zheng[6], Yan Shi[7], Xiaolin Sun[8], Zhanguo Li[8], Chunyang Xiong[2,9], Wanli Liu[1]*

[1]MOE Key Laboratory of Protein Sciences, Collaborative Innovation Center for Diagnosis and Treatment of Infectious Diseases, School of Life Sciences, Institute for Immunology, Tsinghua University, Beijing, China; [2]Academy for Advanced Interdisciplinary Studies, Peking University, Beijing, China; [3]School of Medicine, Zhejiang University, Hangzhou, China; [4]Chengdu Institute of Biology, Chinese Academy of Sciences, Chengdu, China; [5]State Key Joint Laboratory of Environment Simulation and Pollution Control, School of Environment, Tsinghua University, Beijing, China; [6]Department of Rheumatology and Clinical Immunology, Peking Union Medical College Hospital, Peking Union Medical College and Chinese Academy of Medical Sciences, Beijing, China; [7]Center for Life Sciences, Department of Basic Medical Sciences, Institute of Immunology, Tsinghua University, Beijing, China; [8]Department of Rheumatology and Immunology, Clinical Immunology Center, Peking University People's Hospital, Beijing, China; [9]College of Engineering, Peking University, Beijing, China

*For correspondence: liuwanli@biomed.tsinghua.edu.cn

†These authors contributed equally to this work

Competing interests: The authors declare that no competing interests exist.

**Abstract** The mechanosensing ability of lymphocytes regulates their activation in response to antigen stimulation, but the underlying mechanism remains unexplored. Here, we report that B cell mechanosensing-governed activation requires BCR signaling molecules. PMA-induced activation of PKCβ can bypass the Btk and PLC-γ2 signaling molecules that are usually required for B cells to discriminate substrate stiffness. Instead, PKCβ-dependent activation of FAK is required, leading to FAK-mediated potentiation of B cell spreading and adhesion responses. FAK inactivation or deficiency impaired B cell discrimination of substrate stiffness. Conversely, adhesion molecules greatly enhanced this capability of B cells. Lastly, B cells derived from rheumatoid arthritis (RA) patients exhibited an altered BCR response to substrate stiffness in comparison with healthy controls. These results provide a molecular explanation of how initiation of B cell activation discriminates substrate stiffness through a PKCβ-mediated FAK activation dependent manner.

## Introduction

Cells interact with an intricate mechanical extracellular microenvironment by sensing physical signals and using surface receptors to convert them into chemical signals and, subsequently, cellular responses (*Sun et al., 2012*; *Liu et al., 2015*). Lymphocytes – the cells of the adaptive immune system – include B and T cells, both of which use surface expressed antigen receptors to sense external

**eLife digest** The human immune system protects the body from harmful bacteria, viruses and other microbes. Immune cells called B cells use proteins called B cell receptors on their surface to identify these invaders. When the B cell receptors detect molecules called antigens on the surface of the microbes, they produce signals that activate the B cell and enable it to combat the infection.

Previous research has found that B cells react differently to antigens depending on the stiffness of the surface to which an antigen is attached. Now, Shaheen, Wan, Li et al. have attached antigens to artificial surfaces that were either stiff or soft and examined how B cells responded to them. Some of the B cells were modified to lack particular molecules that are important for B cell receptor signaling.

The results of the experiments suggest that two signaling molecules – called protein kinase C beta (PKC$\beta$) and focal adhesion kinase (FAK) – enable B cells to distinguish between the stiffness of different surfaces. PKC$\beta$ activates FAK, which causes the B cell to spread onto the surface and stick to it. However, B cells that had an inactive version of FAK – or lacked the protein entirely – did not efficiently spread onto the surfaces and were less able to discriminate between stiff and soft surfaces.

In autoimmune diseases such as rheumatoid arthritis, B cells are overactive and attack the body's own cells. Shaheen et al. found that the B cells of people with rheumatoid arthritis are less able to distinguish between stiff and soft surfaces than normal cells. Further research that investigates how to change the ability of a B cell to detect stiffness could therefore help researchers to develop treatments or vaccines for rheumatoid arthritis and other autoimmune conditions.

antigens. The B cell receptor (BCR) is composed of a membrane-bound entity of immunoglobulin (Ig) and an Ig$\alpha$-Ig$\beta$ heterodimer in a 1:1 stoichiometry (*Pierce and Liu, 2010*). The engagement of the BCR and antigen leads to initiation of B cell activation. It is known that antigens exhibit great diversity and B cell activation is remarkably sensitive to the diversity of antigen properties, including antigen density (*Liu et al., 2010a*; *Fleire et al., 2006*), antigen affinity (*Liu et al., 2010a*; *Fleire et al., 2006*), antigen valency (*Bachmann et al., 1993*; *Liu and Chen, 2005*; *Liu et al., 2004*), and Brownian mobility of the antigen (*Wan and Liu, 2012*). Previous studies have suggested the extraordinary capability of B cells to sense the chemical and physical features of the antigen, although the presenting forms of antigen are far more complicated under physiological conditions. Recent studies have also suggested that the antigens encountered by B cells in vivo are presented on substrates with various stiffness (also referred as rigidity or elasticity) features (*Bachmann and Jennings, 2010*). Stiffness describes the extent that an object resists deformations in response to an applied force, which is quantified by Young's modulus in units of Pascal (Pa or N/m$^2$ or m$^{-1}$·kg·s$^{-2}$) (*Discher et al., 2005*). For example, antigen molecules presented by a viral capsid exhibit a high degree of stiffness (45,000–1,000,000 kPa) (*Mateu, 2012*), while the same antigen molecules expressed on the membrane of a virus-infected host cell show a medium level of stiffness (0.01–1000 kPa) (*Nemir and West, 2010*), and if released from the virus into the plasma display a particularly low degree of stiffness (several Pa) (*Araujo et al., 2012*).

B cells acquire extracellular matrix(ECM)-associated antigens in a B cell receptor (BCR) and contact-dependent manner (*Ciechomska et al., 2014*), in the presence of a considerable range of stiffness of ECM in the tissue, from 0.012 kPa to 20 kPa (*Bao and Suresh, 2003*; *Paszek et al., 2005*; *Engler et al., 2004*; *Nemir and West, 2010*). Recent studies have shown that the degree of stiffness of the substrates presenting the antigens efficiently regulates the activation of both B and T cells (*Wan et al., 2013*; *Bashour et al., 2014*; *O'Connor et al., 2012*; *Judokusumo et al., 2012*; *Zeng et al., 2015*; *Wan et al., 2015*; *Saitakis et al., 2017*); however, the underlying molecular mechanism remains unexplored.

The ECM-associated microenvironment is an abundant source of antigens (*Tesniere et al., 2008*; *Schaefer, 2010*; *Knight, 2015*). The capability of B cells to sense the stiffness features of antigen-presenting surfaces may be related to the pathological activation of auto-reactive B cells in autoimmune disease patients. For example, in patients with rheumatoid arthritis (RA), reduced cartilage

stiffness leads to auto-antigen-presenting B cells, which subsequently causes the production of auto-antibodies (*Mauri and Ehrenstein, 2007*). Clinical reports strongly suggest that the altered stiffness properties of the ECM are linked to aberrant activation of auto-antigen reactive B cells. However, it is unknown whether the primary B cells in RA patients maintain their capability to discriminate the stiffness of the substrates presenting antigens, and if they do, how such capabilities differ from those of primary B cells derived from healthy individuals. Canonically, it is well established that the cells are capable of sensing the mechanical signals from the microenvironment by the conventional mechanosensor that is, integrin. The major types of integrin molecules expressed by B cells are leukocyte function-associated antigen-1 (LFA-1) ($\alpha L\beta 2$) and very late antigen-4 (VLA-4) ($\alpha 4\beta 1$) (*Arana et al., 2008a*). The ligands of LFA-1 and VLA-4 are adhesion molecules that is, intercellular adhesion molecules (ICAM-1/2) and vascular cell adhesion molecule-1 (VCAM-1), respectively (*Arana et al., 2008a*). Previous studies investigating the responses of both B and T cells upon encountering antigens that were tethered to stiff and soft substrates generally did not use adhesion molecules in their experimental systems (*Wan et al., 2013*; *Bashour et al., 2014*; *O'Connor et al., 2012*; *Judokusumo et al., 2012*; *Zeng et al., 2015*). Early studies suggested that lymphocytes discern substrate stiffness independently of integrin signaling, but the details are unclear. More intriguingly, it is also unclear if or how the ability of lymphocytes to discriminate substrate stiffness is regulated by direct interaction between the integrin and adhesion molecule.

Here, we addressed all three of the aforementioned questions through a combination of molecular imaging with genetic and pharmacological approaches by examining initiation of B cell activation on antigen-presenting substrates with stiff or soft features. The two most commonly used substrates for in vitro mechanosensing studies are polyacrylamide (PA) and polydimethylsiloxane (PDMS). Both were used in this study to fabricate the soft and stiff substrates at contrasting ranges of 2.6 versus 22.1 kPa for PA (*Wan et al., 2013*; *Judokusumo et al., 2012*) and 20 versus 1100 kPa for PDMS (*O'Connor et al., 2012*). The results revealed that BCR signaling competent B cells can discriminate substrate stiffness through accumulating/polarizing more BCR molecules into the B cell immunological synapse (IS) when encountering stiff rather than soft substrates. In contrast, B cells deficient for each of the early BCR signaling molecules including tyrosine-protein kinase (Lyn), spleen tyrosine kinase (Syk), phospholipase C$\gamma$2 (PLC$\gamma$2), Bruton's tyrosine kinase (Btk), B-cell linker protein (BLNK), and protein kinase C (PKC$\beta$), lose this discrimination capability. Moreover, we found that PKC$\beta$ functions downstream of Btk and PLC$\gamma$2, as PMA-induced activation of PKC$\beta$ can bypass the requirements of Btk and PLC$\gamma$2 for B cell discrimination of substrate stiffness. Mechanistically, we excluded PKC$\beta$-mediated NF-$\kappa$B activation in the capability of B cells to discriminate substrate stiffness. Instead, we provide evidence that PKC$\beta$-dependent activation of FAK is required for B cells to discriminate substrate stiffness, whereby FAK potentiates B cell spreading and adhesion responses. As supporting evidence for this model, we observed that a pharmaceutical inhibitor targeting FAK, the key downstream molecule in integrin signaling, drastically impaired the capability of B cells to discriminate substrate stiffness. FAK-deficient B cells also lost this capacity to discriminate substrate stiffness, which was rescued by exogenous expression of FAK-WT but not the inactivated mutant FAK-Y926F. Further data showed that the 'outside-in' activation of integrin by the adhesion molecules, ICAM-1 or VCAM-1, greatly enhanced the B cell's capability to discriminate substrate stiffness. Lastly, the capability of B cells to discriminate substrate stiffness could be readily recapitulated in human primary PBMC B cells. It was striking to observe that B cells from RA patients exhibited a disordered and weakened capability to discriminate substrate stiffness in comparison with the healthy controls. Our data dissect the molecular mechanisms whereby B cells discriminate substrate stiffness during B cell activation, and also explicitly determine the contribution of adhesion molecules to such an event. The conclusion that B cells discriminate substrate stiffness through a PKC$\beta$-mediated FAK activation-dependent manner in the initiation of B cell activation improves our understanding of the sophisticated mechanosensing capability of B lymphocytes and provides a potential explanation for the dysregulated activation of auto-reactive B cells in RA patients.

## Results

### Use of PDMS/PA substrates for the molecular dissection of B cell stiffness discrimination

To dissect the molecular mechanism of substrate stiffness discrimination by B cells, we used a library of DT40 B cells deficient for specific signaling molecules through a gene-targeted knock out (KO) technique (*Kurosaki et al., 2010*; *Kurosaki, 1999*). The KO library of DT40 B cells was established by Kurosaki and his colleagues to investigate the function of specific signaling molecules in BCR signal transduction (*Kurosaki et al., 2010*; *Kurosaki, 1999*). To determine whether DT40 wild-type (DT40-WT) B cells can discriminate between stiff and soft PDMS substrates during initiation of B cell activation, we used mouse anti-chicken IgM antibodies that were pre-tethered to stiff or soft PDMS substrates as a surrogate antigen to activate DT40 B cells. As antigen density can drastically influence B cell activation (*Liu et al., 2010a*; *Fleire et al., 2006*), we first examined the distribution and density of the surrogate antigen on both stiff and soft PDMS using Alexa Fluor 647-conjugated mouse IgM antibody (clone M4) and anti-chicken IgM as surrogate antigens. We measured an even distribution of the antigens on both types of PDMS substrates through confocal fluorescence microscopy (*Figure 1A,B*). We confirmed that there was comparable accessibility to the non-fluorophore conjugated surrogate antigens on both of the substrates using the DyLight 649-conjugated Fab anti-mouse IgM antibody (*Figure 1C,D*). To avoid inadvertently altered antigen accessibility for B cells when varying the stiffness of PDMS substrates, we examined the accessibility of the antigens to DT40-WT B cells. We set equal amounts of DT40-WT B cells on either stiff or soft PDMS substrates that were pre-coated with the same density of surrogate antigens. We counted the number of tethered DT40-WT cells under the microscope and found that DT40-WT B cells can equally tether to both stiff and soft PDMS substrates (*Figure 1E–G*). Thus, we established an experimental system constituting stiff and soft PDMS substrates that presented the same density of antigen with equal accessibility to DT40 B cells.

Next, we compared the capability of DT40-WT B cells to discriminate substrate stiffness during their activation initiation by quantifying the amount of BCRs that accumulated at the contact interface between B cells and the antigen-presenting surfaces on either soft or stiff substrates (*Figure 2A,B*). BCRs are evenly distributed in quiescent B cells, and the strength of the initiation of B cell activation can be measured by the level of polarization of the BCRs to the site of contact with the antigen-presenting surfaces in activated B cells (*Liu et al., 2010b*, *2010c*, *2012*; *Seeley-Fallen et al., 2014*; *Liu et al., 2013*; *Arana et al., 2008b*; *Carrasco and Batista, 2007*; *Lin et al., 2008*; *Treanor et al., 2011*; *Weber et al., 2008*; *Depoil et al., 2008*; *Fleire et al., 2006*). To quantify the amount of accumulated BCRs, we used the mean fluorescence intensity (MFI) of BCRs within the B cell contact interface rather than the total fluorescent intensity (TFI) value, as the latter will increase in response to B cell spreading during B cell activation, which increases the size of the contact interface. Thus, it is not possible to distinguish whether the increase of TFI results from polarization of BCRs to the B cell contact interface or from an increase in the size of the contact interface. In contrast, the value of MFI is resilient to such changes as MFI is defined by a value of TFI / size of the contact interface, equal to the density of BCRs within the contact interface, a change that can only be introduced by the enrichment of BCRs. Indeed, the results showed a much higher BCR MFI in B cells that were placed on stiff substrates compared with B cells on soft substrates (*Figure 2B*). To better compare the efficiency of the accumulation of BCRs at the B cell's contact interface with either stiff or soft PDMS substrates, we defined a ratio index as the BCR MFI of each cell on the stiff substrate divided by the averaged BCR MFI value of all cells on the soft substrate. A ratio larger than 1 would indicate that B cells can accumulate more BCRs when on a stiff substrate versus a soft substrate, and a higher ratio value would indicate better discrimination capability. Another advantage of using such a ratio is to enable multi-grouped comparisons, which are problematic for absolute MFI values because of the presence of inter-sample and inter-batch variations. Using this approach with DT40-WT B cells, we found the ratio of the MFI of BCR on stiff/soft PDMS substrates was larger than 1.5, suggesting that stiff substrates induced the accumulation of significantly more BCRs into the B cell IS compared with soft substrates (*Figure 2B*). Thus, DT40-WT B cells could clearly discriminate between stiff and soft PDMS substrates (*Figure 2A,B*). Similar results were acquired in the same experimental system using PA substrates (*Figure 2C,D*). These results validate the utility

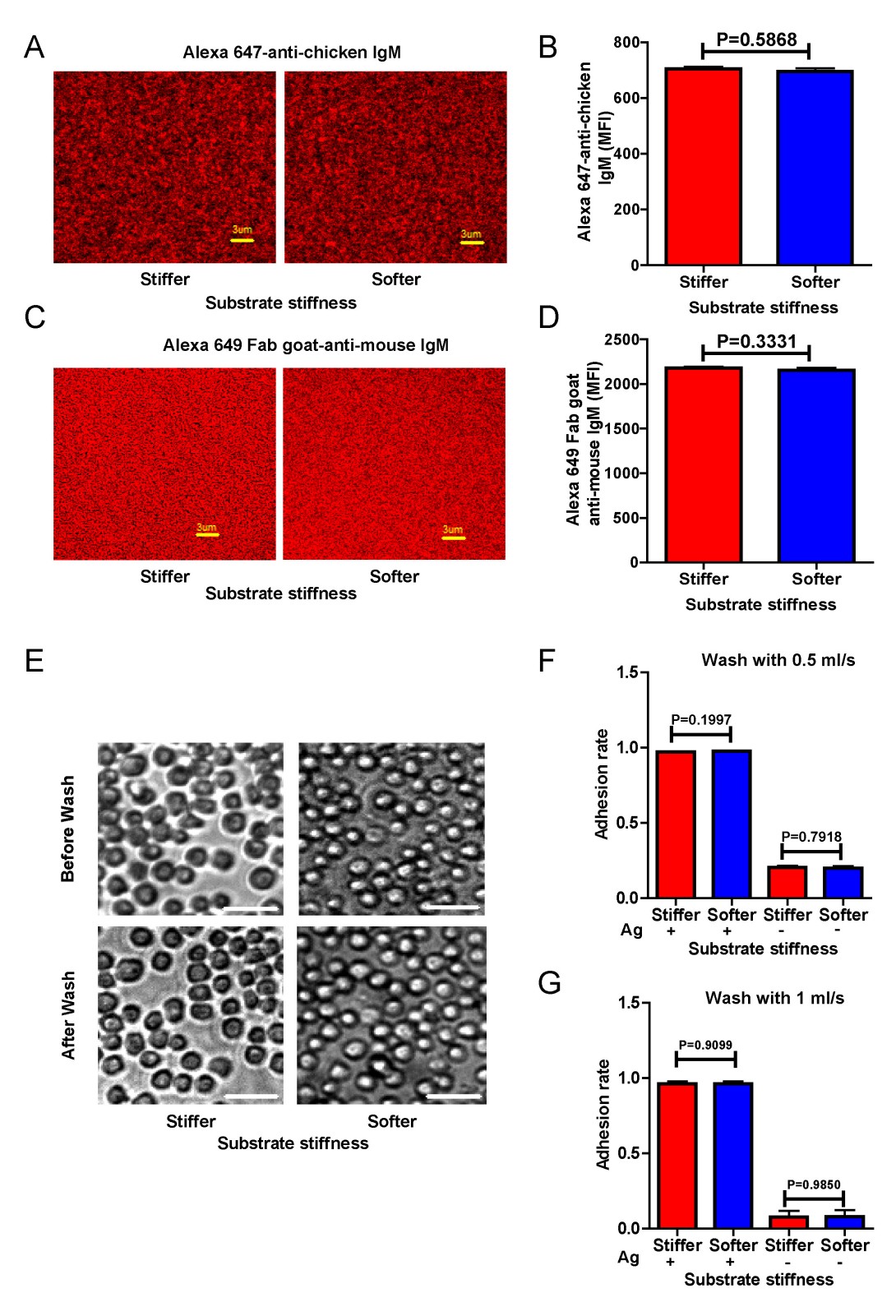

**Figure 1.** Surrogate antigens tethered to stiff or soft PDMS substrate show similar surface density and accessibility for B cells. (**A**) Distribution of Alexa 647-conjugated mouse IgM monoclonal antibody (clone M4) anti-chicken IgM as a surrogate antigen on the surface of PDMS substrates was equal and highly uniform as shown by confocal fluorescence microscopy. Scale bar is 3 µm. (**B**) Quantification of antigen density on both the surfaces of stiff and soft PDMS. (**C**) Representative confocal fluorescence microscope images showing the equal and highly uniform accessibility of the surrogate antigen on

*Figure 1 continued on next page*

*Figure 1 continued*

both the substrates as probed by the DyLight 649-conjugated Fab anti-mouse IgM antibody. Scale bar is 3 µm. (**D**) Quantification of antibody accessibility on both the surfaces of stiff and soft PDMS. Shown are mean ± SEM from one representative of three independent experiments. Two-tailed *t* tests were performed for statistical comparisons. (**E**) Representative images of the adhesion of DT40 B cells on the surface of either stiff or soft PDMS substrates before and after wash with 10 ml of PBS-1% FBS washing buffer. Scale bar is 50 µm. (**F, G**) Statistical quantification of the percentage of DT40 B cells adhered to stiff or soft substrates with or without tethered antigens. Adhesion rate is used for quantification as detailed in Materials and methods. The results were obtained using two different washing speeds of 0.5 (**F**) or 1 ml/sec (**G**) for a total amount of 10 ml of PBS-1% FBS washing buffer. Bar represents mean ± SEM from one representative of two independent experiments. Two-tailed *t* tests were performed for statistical comparisons.

of using DT40 B cells in this PDMS or PA based experimental system for dissecting the molecule mechanisms underlying the capability of B cells to discriminate substrate stiffness during the initiation of B cell activation.

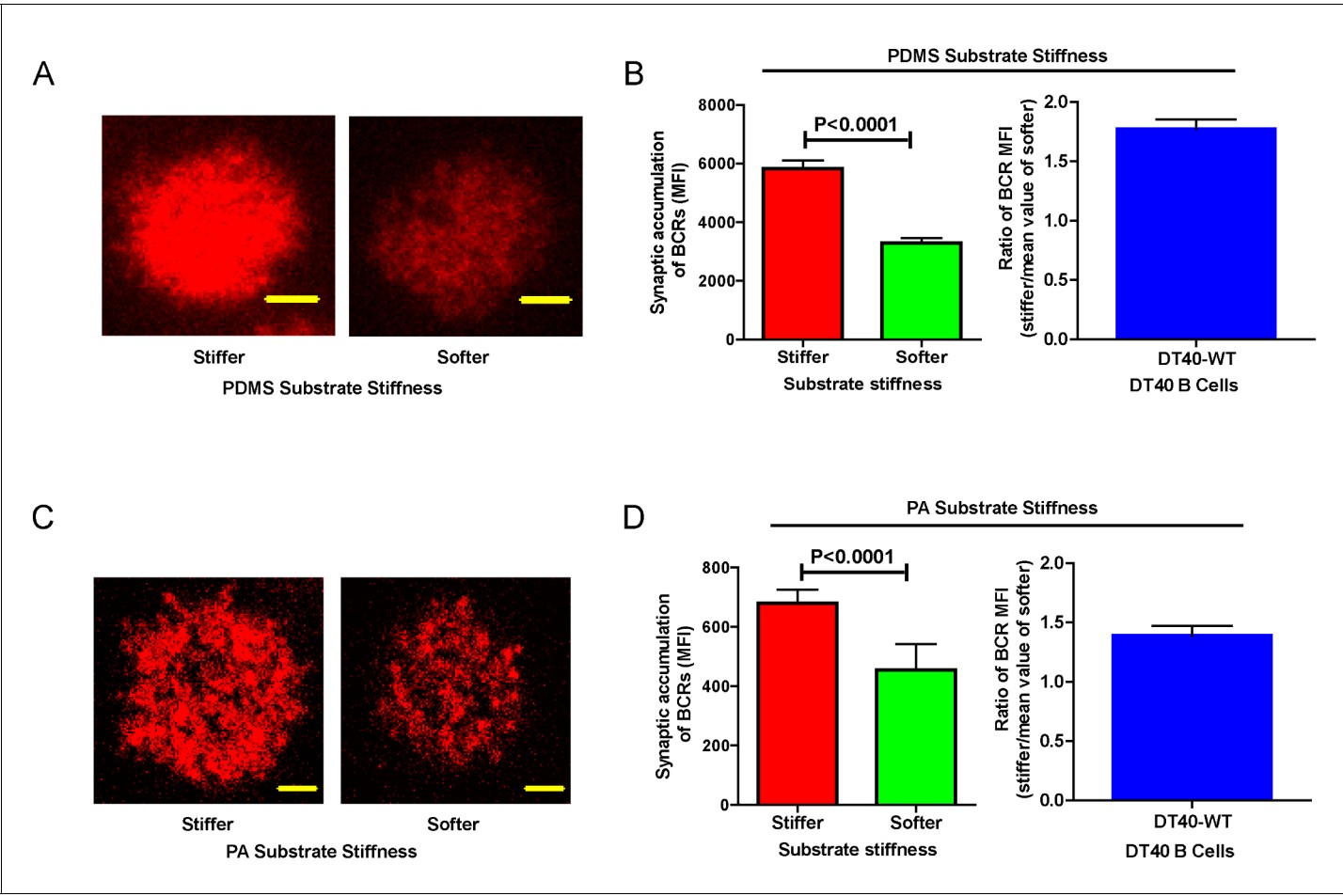

**Figure 2.** DT40-WT B cells exhibit excellent capability to discriminate substrate stiffness. (**A**) Representative confocal images of DT40 B cells showing the contact interface with the antigens tethered on either stiff or soft PDMS substrates. Scale bar is 3 µm. (**B**) Synaptic accumulation of BCRs on either stiff or soft substrates and a ratio figure showing the BCR MFI of each cell on stiff substrates to the averaged value of the BCR MFI of all the cells on soft PDMS substrates. (**C**) Representative confocal images of DT40 B cells showing the contact interface with the antigens tethered on either stiff or soft PA substrates. Scale bar is 5 µm. (**D**) Synaptic accumulation of BCRs on either stiff or soft substrates and a ratio figure showing the BCR MFI of each cell on stiff substrates to the averaged value of the BCR MFI of all the cells on soft PA substrates. Bar represents mean ± SEM from one representative of two independent experiments. Two-tailed *t* tests were performed for statistical comparisons.

## Lyn, Syk, and PLCγ2 are required for the substrate stiffness discrimination capability of B cells

We first performed experiments in DT40 B cells by knocking out the key molecules in BCR signaling, including Lyn (*Takata et al., 1994*), Syk (*Takata et al., 1994*), PLCγ2 (*Watanabe et al., 2001*), Btk (*Hashimoto et al., 1999*), BLNK (*Ishiai et al., 1999*), and PKCβ (*Shinohoto et al., 2005*). Lyn, Syk, and PLCγ2 are well characterized signaling molecules that function as an initiator complex to trigger the BCR signaling cascade, leading to subsequent assembly of a multi-protein complex of adaptors and various signaling molecules proximal to the plasma membrane, known as the signalosome (*Kurosaki, 1999*; *Kurosaki et al., 2010*). To reduce batch to batch variation, each of the three types of DT40 KO was respectively assayed in parallel with the corresponding DT40-WT B cells. In each case, DT40-WT B cells showed a clear capability to discriminate substrate stiffness with a ratio of approximately 1.5. However, the DT40-Lyn-KO, DT40-Syk-KO, and DT40-PLCγ2-KO B cells only showed a ratio of 1.0, suggesting that similar amounts of BCRs accumulated at the contact interface of B cells and the PDMS substrates, regardless of stiffness (*Figure 3A–C*). Similar results were obtained using PA substrates (*Figure 3—figure supplement 1A–C*). As the synaptic accumulation of BCRs in these three types of KO B cells was largely BCR signaling-independent, these results indicate that only BCR signaling-dependent enrichment of BCRs to the B cell IS is regulated by the stiffness feature of the substrates.

To verify that these observations were dependent on the knocked out molecules rather than subordinate effects during the construction of these three types of KO B cells, we cloned the cDNA of Lyn, Syk, and PLCγ2 molecules by RT-PCR from the mRNA of DT40-WT B cells (*Table 1*) and performed rescue experiments (*Figure 3A–C*). Importantly, exogenous overexpression of these molecules rescued the capability of B cells to discriminate substrate stiffness, suggesting that the observed phenotype is indeed mediated through these signaling molecules. To further validate these observations in other types of B cells including primary B cells, we used pharmaceutical inhibitors specifically targeting each of these three signaling molecules. Pharmaceutical inhibitor PP2, Piceatannol, and U73122 were used to block the function of Lyn, Syk, and PLCγ2 molecules, respectively, as detailed in the Materials and methods section. Hapten antigen 4-hydroxy-3-nitrophenyl acetyl (NP)-specific B1-8 primary B cells from B1-8 transgenic mice (*Hauser et al., 2007*) or B lymphoma cells, CH27 that were pre-treated with each of these inhibitors lost the capacity to discriminate between the different degrees of stiffness of the substrate-presenting antigens (*Figure 3D–I*). We next examined whether this loss of ability to discriminate substrate stiffness resulted from changes in the BCR MFI (as a parameter indicating BCR microclustering) on a stiff or soft substrate surface. We achieved this by comparing the BCR MFI of the inhibitor pre-treated versus DMSO-control pre-treated primary B cells (*Figure 3—figure supplement 1D–F*) on the surface of the substrates with the same Young's modulus (stiffness). The results demonstrated that inhibition of proximal signaling molecules, Lyn, Syk, or PLCγ2 blocked the synaptic accumulation of BCRs specifically on stiff substrates. In comparison, we found only mild differences in BCR MFI of primary B cells pre-treated with inhibitor versus DMSO-control on soft substrates (*Figure 3—figure supplement 1D–F*). Thus, BCR signaling initiating molecules, Lyn, Syk, and PLCγ2 are required for B cells to discriminate between stiff and soft substrates during the initiation of B cell activation.

## Genetic ablation of Btk, BLNK, or PKCβ blunts the ability of B cell to discriminate substrate stiffness

We further assessed the contribution of downstream BCR signaling molecules in B cell discrimination of substrate stiffness by examining DT40 B cells deficient for Btk, BLNK, or PKCβ. We first observed that DT40-Btk-KO, DT40-BLNK-KO, and DT40-PKCβ-KO B cells encountering antigens presented on soft substrates showed comparable accumulation of BCRs into the B cell IS as DT40-WT B cells (*Figure 4A–C*). These results are consistent with an earlier report showing that DT40-Btk-KO, DT40-BLNK-KO, and DT40-PKCβ-KO B cells accumulated 81%, 98%, and 82%, respectively, of the membrane-bound antigens on fluid planar lipid bilayer membranes to the B cell IS compared with DT40-WT B cells (*Weber et al., 2008*). However, when quantifying the substrate stiffness discrimination capability of these KO B cells on substrates with different stiffness features, it was striking to observe that each of these three B cell lines completely lost the capability to discriminate substrate stiffness (*Figure 4D–F*). Similar results were obtained using the PA gel system (*Figure 4—figure supplement*

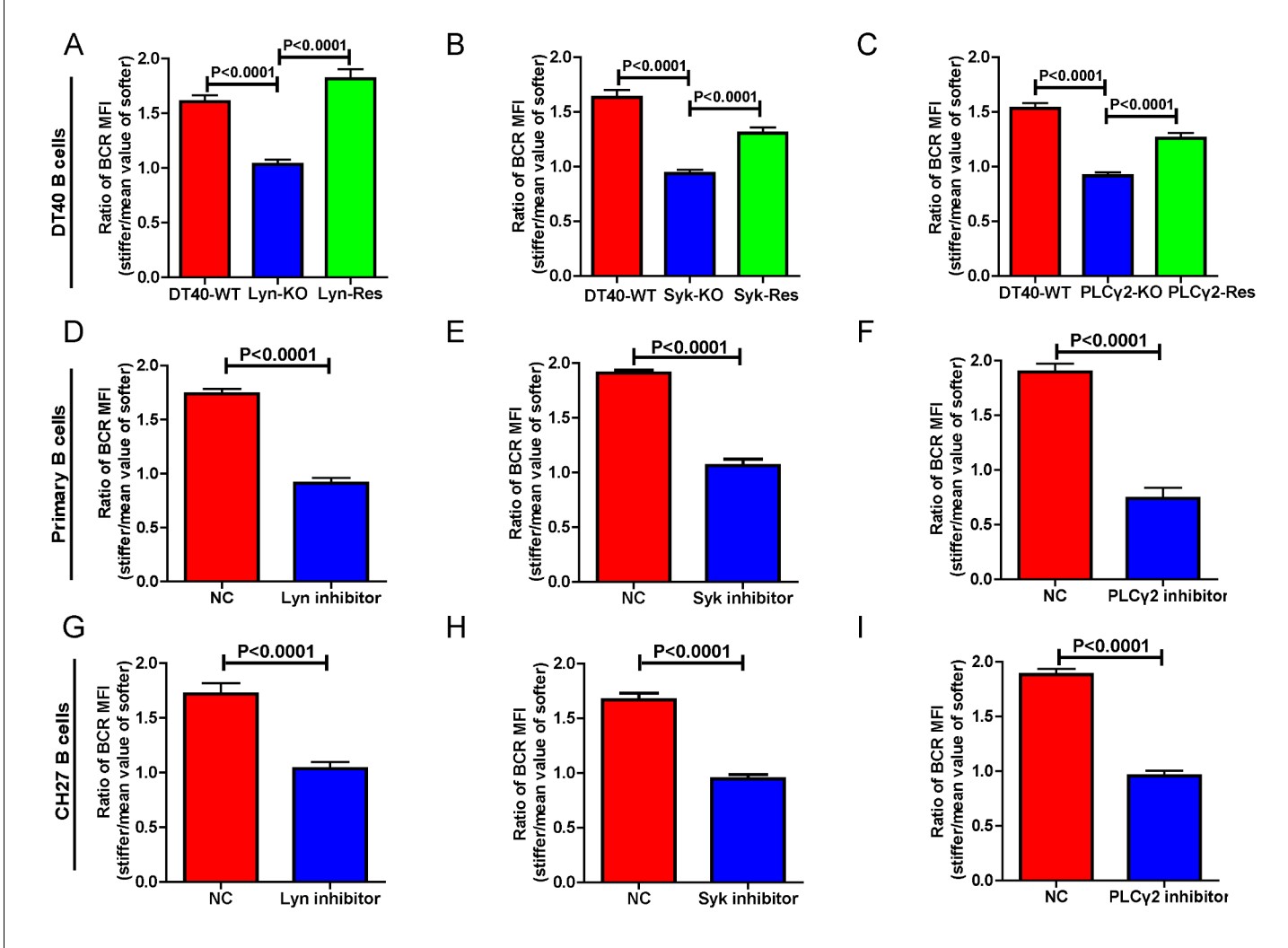

**Figure 3.** Lyn, Syk, and PLCγ2 are required for B cells to discriminate substrate stiffness. (A–C) Statistical comparison for the substrate stiffness discrimination capability of B cells from the following three groups: (A) DT40-WT, DT40-Lyn-KO, and DT40-Lyn-Rescue cells; (B) DT40-WT, DT40-Syk-KO, and DT40-Syk-Rescue cells; (C) DT40-WT, DT40-PLCγ2-KO, and DT40-PLCγ2-Rescue cells. The calculation of ratio of BCR MFI is defined as the BCR MFI of each cell on stiff substrates to the averaged value of the BCR MFI of all the cells on soft substrates. (D–F) B1-8 primary B cell pre-treatment with DMSO as a control (NC) versus Lyn inhibitor PP2 (D), Syk inhibitor Piceatannol (E), and PLCγ2 inhibitor U73122 (E). (G–I) CH27 B cell pre-treatment with DMSO as a control (NC) versus Lyn inhibitor PP2 (G), Syk inhibitor Piceatannol (H) and PLCγ2 inhibitor U73122 (I). In (A) to (I), bar represents mean ± SEM from at least 20 cells in one representative of three independent experiments. Two-tailed *t* tests were performed for statistical comparisons.

The following figure supplement is available for figure 3:

**Figure supplement 1.** Lyn, Syk, PLCγ2, Btk, BLNK, or PKCβ molecules are required for B cells to discriminate between stiff and soft substrates.

*1A–C*). Exogenous supply of Btk, BLNK, or PKCβ rescued the capability of B cells to discriminate substrate stiffness (*Figure 4D–F*). Furthermore, the deficiency of each of these three proximal signaling molecules blocked the synaptic accumulation of BCRs specifically on stiff substrates, while there were only minor changes in BCR MFI on soft substrates (*Figure 4—figure supplement 1D–F*). As Btk, BLNK, and PKCβ are all well-established signaling molecules in BCR signal transduction, our data also demonstrated that the initiation of B cell activation is sensitive to substrate stiffness in a BCR signaling-dependent manner.

**Table 1.** Primer sequences used to amplify cDNA

| Gene name | Upstream primer | Downstream primer |
|---|---|---|
| Lyn | atgggatgtataaaatcaaaaagga | ctatggctgctgttgatattgcc |
| Syk | atggcttccaacatggccaacc | aatcacccttttacagcattatcatcaaggcatt |
| PLCγ2 | atgcctcgaaagagtgtagattatg | ttaagagtagaatttgctgttactg |
| Btk | atggccagcatcatcctg | tcacggctcttcgtctg |
| PKCβ | gcctaccccaagtccatgt | cttggtcatgagccctttg |
| FAK | ggcagcagcttaccttgatcc | ggcctggactggctgatcatt |
| BLNK | gcggccgcaccggtctgcagctagctggacaagctgaataagataactg | ctcaccatcgaagcagctcctcctcctgaaaccttcacagcatatttcagtc |

## PMA-induced activation of PKCβ can bypass the requirement of Btk and PLCγ2 for substrate stiffness discrimination by B cells

In the sequential initiation and transduction of the membrane proximal BCR signaling cascade, antigen stimulation of BCR leads to phosphorylation of the immunoreceptor tyrosine-based activation

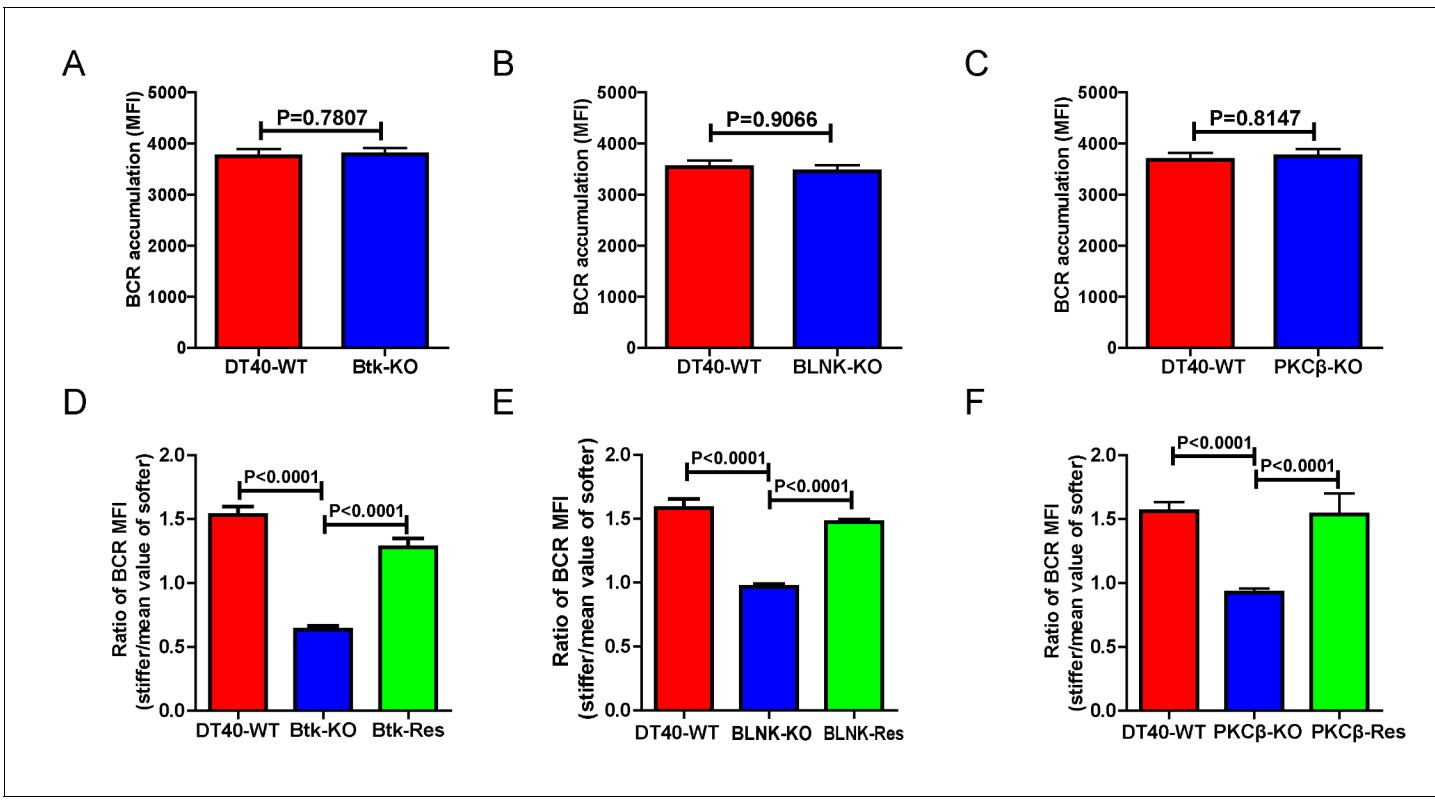

**Figure 4.** Genetic ablation of Btk, BLNK, or PKCβ blunts the ability of B cells to discriminate substrate stiffness. (A–C) B cells showed comparable capability to accumulate BCRs into B cell IS in the following groups (A) DT40-WT and DT40-Btk-KO; (B) DT40-WT and DT40-BLNK-KO; and (C) DT40-WT, DT40-PKCβ-KO. (D–F) Statistical comparison for the substrate stiffness discrimination capability of B cells from the following three groups: (D) DT40-WT, DT40-Btk-KO, and DT40-Btk-Rescue cells; (E) DT40-WT, DT40-BLNK-KO, and DT40-BLNK-Rescue cells; (F) DT40-WT, DT40-PKCβ-KO, and DT40-PKCβ-Rescue cells. In (A)-(F), bar represents mean ± SEM from at least 27 cells in one representative of three independent experiments. Two-tailed *t* tests were performed for statistical comparisons.

The following figure supplement is available for figure 4:

**Figure supplement 1.** Genetic ablation of Btk, BLNK, or PKCβ blunts the ability of B cells to discriminate substrate stiffness.

motif (ITAM) in the cytoplasmic domains of BCR components Igα/Igβ by Lyn kinase. The phosphorylated ITAMs provide a docking site for Syk kinase, which results in rapid auto-phosphorylation of Syk within its linker regions (*Saouaf et al., 1994*), which subsequently provides the additional docking sites for PLCγ2 (*Weber et al., 2008*) and Btk (*Smith et al., 1994*). Btk actively promotes the phosphorylation and activation of PLCγ2. Activated PLCγ2 potently hydrolyzes phosphatidylinositol 4,5-biphosphate to generate inositol 1,4,5-trisphosphate ($IP_3$) and diacylglycerol (DAG). DAG recruits PKCβ to the membrane proximal BCR signalosome.

PMA, a phorbol ester, has been widely used as a DAG analog to directly induce activation of PKC. Thus, we tested whether addition of PMA could rescue the inability of DT40-BTK-KO, DT40-PLCγ2-KO, and DT40-PKCβ-KO B cells to discriminate substrate stiffness. We chose these three types of B cell lines considering that DT40-BTK-KO and DT40-PLCγ2-KO B cells lack the ability to produce DAG upon BCR activation while maintaining normal expression of PKCβ. In contrast, DT40-PKCβ-KO can normally produce DAG upon BCR activation, but lacks PKCβ. In our experimental system, we pre-treated either WT or KO DT40 B cells with different concentrations of PMA, 5, 20, and 50 ng/ml, following a published protocol (*Quann et al., 2009*), and then examined the substrate stiffness discrimination capability of B cells under each of these conditions. DT40-WT B cells showed good substrate stiffness discrimination capability (ratio = 1.5) in the absence of PMA as expected, and this capability was largely unaffected in the presence of PMA (*Figure 5A–D*). In contrast, PMA rescued the ability of DT40-PLCγ2-KO, DT40-BTK-KO, and DT40-BLNK-KO B cells to discriminate substrate stiffness in a dose-dependent manner. This suggests that PMA-mediated activation of PKCβ that bypasses early BCR signaling molecules is sufficient to restore the substrate stiffness discrimination capability of B cells (*Figure 5A–C*). In contrast, PMA-treated DT40-PKCβ-KO B cells failed to exhibit such discrimination capability (*Figure 5D*). A much higher concentration of PMA (100 ng/ml) still failed to rescue the defect in DT40-PKCβ-KO B cells (*Figure 5E*). DT40-WT B cells pre-treated with a PKCβ inhibitor and PMA also failed to discriminate stiffness (*Figure 5F*). Together, these results suggest that Btk and PLCγ2 function upstream of PKCβ, demonstrating the key role of PKCβ in mediating the capability of B cells to discriminate substrate stiffness.

## PKCβ-dependent FAK activation accounts for B cell discrimination against substrate stiffness

Next, we investigated the requirement for BCR signaling molecules by B cells to discriminate substrate stiffness. PKCβ is the most downstream signaling molecule of all those investigated in this report, including Lyn, Syk, PLCγ2, Btk, BLNK, and PKCβ. A canonical function of PKCβ in B cell activation is to directly phosphorylate the downstream signaling molecule Carma-1 at Ser-559, Ser-644, and Ser-652, which induces the association of Carma1 with Bcl10 and Malt1. The formation of Carma1, Bcl10, and Malt1 (CBM) signaling complexes leads to the activation of NF-κB (*Su et al., 2002*; *Shinohara et al., 2005*). However, the activation of NF-κB leads to a downstream gene expression signature in B cells at a late time point, usually hours after the initial BCR and antigen recognition, whereas substrate stiffness discrimination in B cells occurs within a few minutes of interaction between the BCR and antigen. Therefore, we speculated that PKCβ-mediated NF-κB activation would not contribute to the capabilities of B cells to discriminate substrate stiffness. Indeed, we found that Carma1-KO DT-40 B cells effectively maintained their discrimination capability (*Figure 6A*).

The initiation of B cell activation is dependent on adhesion of B cells to the surface of the substrate presenting the antigen. We thus assessed the potential requirement of PKCβ in the discrimination capability of B cells through its function in regulating B cell adhesion responses. Our hypothesis was based on published studies showing that PKC family members play key roles in mediating the adhesion responses of many types of mammalian cells through 'inside-out' activation of integrin molecules (*Haller et al., 1998*; *Besson et al., 2002*; *Disatnik et al., 2002*; *Buensuceso et al., 2005*). Downstream of integrin activation, focal adhesion kinase (FAK), a member of the nonreceptor protein-tyrosine kinase family, is a key player in mechanosensing-mediated cell adhesion responses (*Yu et al., 2012*; *Bashour et al., 2014*; *Slack-Davis et al., 2007*). Moreover, it has also been reported that FAK is required for the chemoattractant-induced migration and adhesion response in B cells (*Tse et al., 2012*). Thus, we first examined the dependence of FAK activation on the substrate stiffness discrimination capability of B cells.

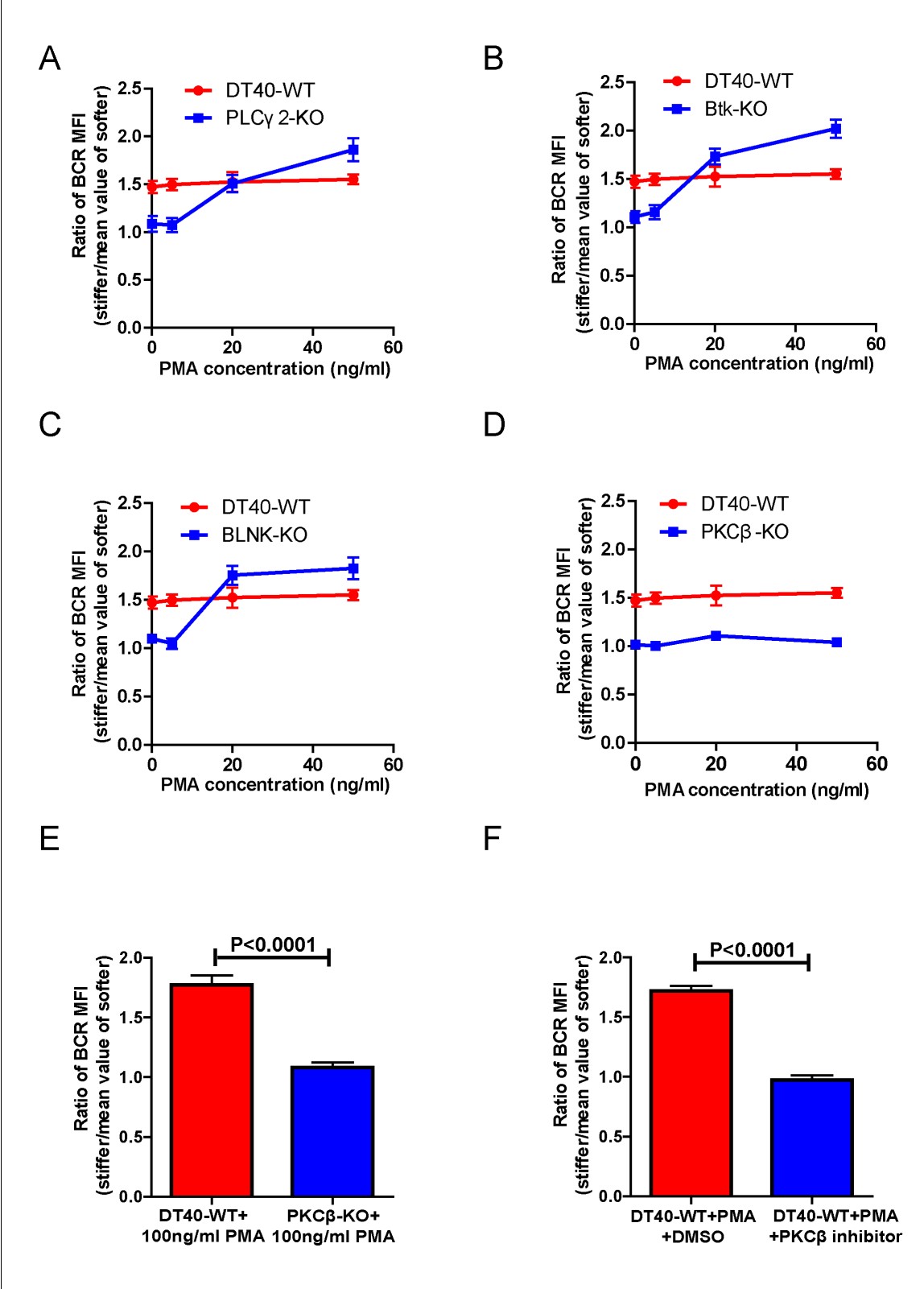

**Figure 5.** PMA-induced activation of PKC$\beta$ can bypass the requirements of Btk and PLC$\gamma$2 for B cells to discriminate substrate stiffness. (A–D) Statistical comparison for the substrate stiffness discrimination capability of PMA pre-treated B cells in the following groups: (A) DT40-WT and DT40-PLC$\gamma$2-KO; (B) DT40-WT and DT40-Btk-KO; (C) DT40-WT and DT40-BLNK-KO; and (D) DT40-WT and DT40- PKC$\beta$-KO. (E) DT40-WT versus DT40-PKC$\beta$-KO B cells that were pre-treated with high concentration of 100 ng/ml PMA. (F) DT40 WT B cells treated with PMA plus DMSO as a control or PMA plus PKC$\beta$

*Figure 5 continued on next page*

*Figure 5 continued*

inhibitor. In (A)-(F), bar represents mean ± SEM from at least 21 cells in one representative of three independent experiments. Two-tailed *t* tests were performed for statistical comparisons.

To test this hypothesis, we used the FAK specific inhibitor, PF573-228 (*Slack-Davis et al., 2007*), to inactivate FAK. Strikingly, PF573-228 pre-treated DT40 B cells, CH27, and B1-8 primary B cells were completely blunted in their capability to discriminate substrate stiffness (*Figure 6B–D*). To confirm this, we knocked out FAK in DT40 B cells (DT40-FAK-KO) using CRISPR/Cas9 gene editing technology (*Figure 6—figure supplement 1A–C*) and found that DT40-FAK-KO B cells lost the capability to discriminate substrate stiffness during their activation. This was rescued by the exogenous expression of chicken FAK-WT (*Figure 6E*). The phosphorylation of FAK at Tyr-925 is considered to be a critical step in FAK activation-mediated migration and adhesion responses (*Deramaudt et al., 2011*). As expected, exogenous expression of a chicken FAK-Y926F mutant (chicken Tyr 926 has sequence homology with Tyr 925 in mouse and human FAK) (*Figure 6—figure supplement 1D*) in DT40-FAK-KO B cells failed to rescue the discrimination capability of B cells (*Figure 6F*). Thus, FAK activation is required for the B cells to discriminate substrate stiffness.

The importance of the phosphorylation of FAK at Tyr-925 (pFAK) in FAK activation-mediated migration and adhesion responses (*Deramaudt et al., 2011*)led us to examine the spatial colocalization of BCR and pFAK molecules within the IS of mouse primary B cells upon activation (*Figure 6G*). As predicted, BCRs primarily aggregated at the central region of the B cell IS, while pFAK accumulated at the peripheral region, consistent with the report that integrin molecules are mainly located at the peripheral region of the B cell IS (*Tolar et al., 2009*). Because we did not use adhesion molecules in our experimental system, we speculated that phosphorylation of FAK would result from activation of BCR signaling pathway molecules upon antigen stimulation. Indeed, as further supporting evidence, we observed no obvious upregulation of pFAK in B cells that were placed on substrates without antigen coating, while very strong pFAK signaling was evident only in the antigen-activated B cells (*Figure 6H*). Strikingly, a significantly stronger pFAK signal was observed in B cells that were placed on stiff substrates than in B cells on soft substrates (*Figure 6I*). Using the method presented above for quantifying the BCR MFI ratio, we also calculated the pFAK MFI ratio index by dividing the pFAK MFI of each cell on a stiff substrate by the averaged value of the pFAK MFI of all cells on the soft substrate (*Figure 6J*). The obtained ratio value of 2.5 for MFI of pFAK on stiff/soft PDMS substrates indicated that stiff substrates induced accumulation of significantly more pFAK molecules into the B cell IS compared with soft substrates (*Figure 6J*). Together, these experiments suggest that the activation of FAK could be induced by antigen-binding-induced BCR signaling in B cells in the absence of adhesion molecules, and more importantly, that FAK activation is required for the B cells to discriminate substrate stiffness.

Based on all these findings, we speculated that activation of FAK by the BCR signaling molecule PKC$\beta$ is important for B cells to discriminate substrate stiffness. To test this, we examined the activation of FAK in DT40-WT and DT40-PKC$\beta$-KO B cells. As expected, the activation of FAK was drastically impaired in DT40-PKC$\beta$-KO cells compared with DT40-WT B cells. Exogenous PKC$\beta$ rescued the capability of B cells to accumulate pFAK in the B cell IS in response to BCR and antigen recognition (*Figure 6K*). As mentioned above, further quantification using the pFAK MFI ratio index demonstrated that DT40-WT B cells exhibited a ratio value of 2.5; in marked contrast, DT40-PKC$\beta$-KO B cells had a ratio of only 1, suggesting that similar amounts of pFAK molecules accumulated at the contact interface of B cells on both stiff and soft PDMS substrates (*Figure 6J*). Furthermore, DT40-PKC$\beta$-KO B cells expressing exogenous PKC$\beta$ accumulated more pFAK molecules on stiff substrates than on soft substrates (*Figure 6J*). As a consequence, exogenous expression of PKC$\beta$ rescued the capability of B cells to discriminate stiffness of the substrates (*Figure 6J*). These results further confirmed the requirement of PKC$\beta$-dependent FAK activation for B cells to effectively discern the stiffness of the substrate.

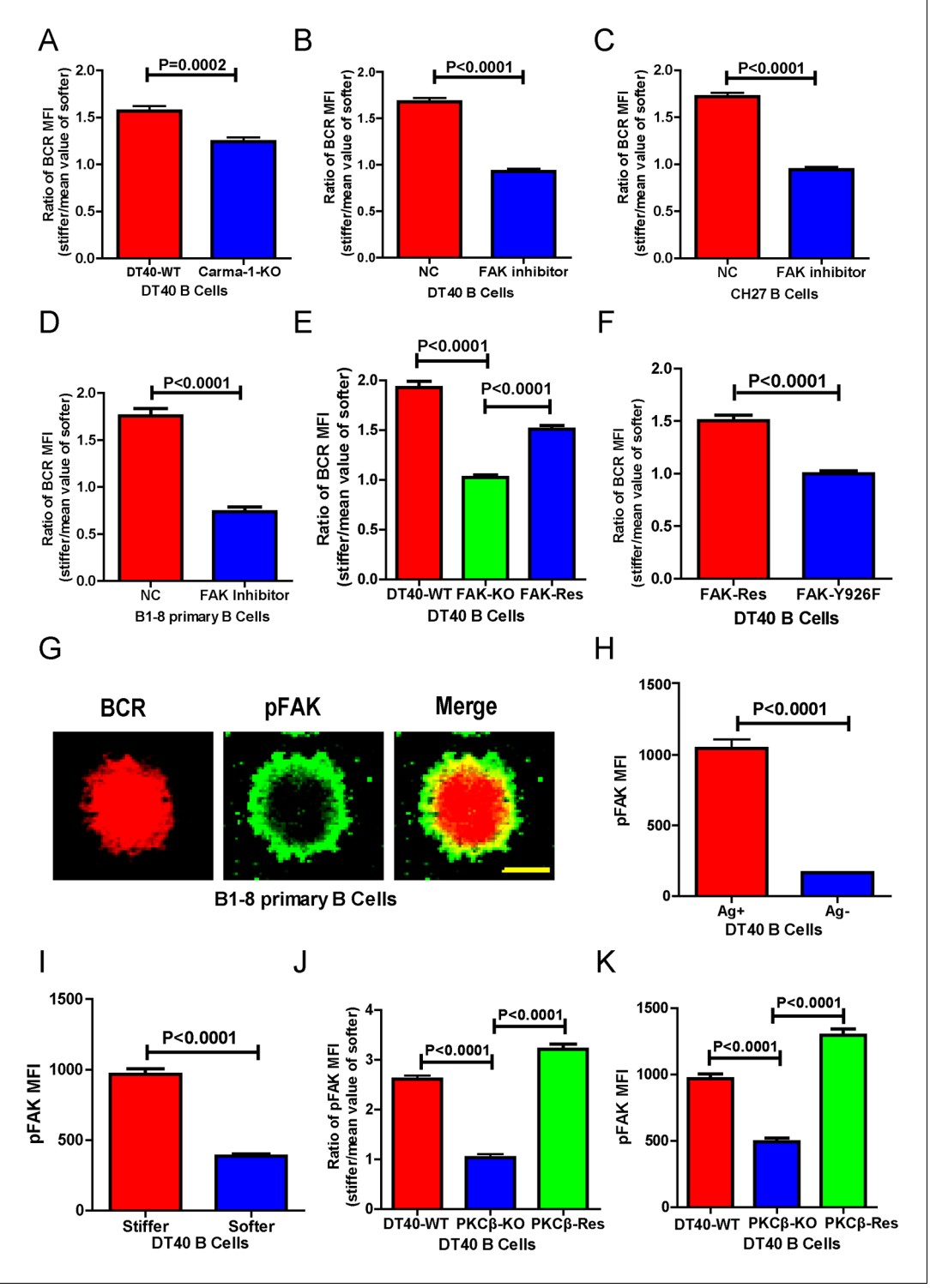

**Figure 6.** PKC$\beta$-dependent FAK activation accounts for B cells to discriminate substrate stiffness. (**A**) Statistical comparison for the substrate stiffness discrimination capability of DT40-WT versus DT40-Carma-1-KO B cells. (**B–D**) Statistical comparison for the substrate stiffness discrimination capability of DT40-WT (**B**), CH27 (**C**), or B1-8 primary (**D**) B cells that were pre-treated with either DMSO as a control (NC) or FAK inhibitor PF573-228 (FAK inhibitor). (**E**) Statistical comparison for the substrate stiffness discrimination capability of DT40-WT, DT40-FAK-KO, and DT40-FAK-Rescue B cells. (**F**) Statistical comparison for the substrate stiffness discrimination capability of DT40-FAK-Rescue and FAK-Y926F Mutant B cells. (**G**) The representative confocal images of B1-8 primary B cells showing the spatial co-distribution of BCR and pFAK (Tyr 925) molecules within the B cell immunological synapse. *Figure 6 continued on next page*

*Figure 6 continued*

Scale bar is 4 μm. (**H**) Statistical comparison of the MFI of pFAK molecules within the B cell immunological synapse of DT40-WT B cells that were placed on the PDMS substrates presenting antigen or lacking antigen. (**I**) Statistical comparison of the MFI of pFAK molecules within the B cell immunological synapse of DT40-WT B cells that were placed on the antigen-presenting surfaces of either stiff or soft PDMS substrates. (**J**) Statistical comparison for the substrate stiffness discrimination capability of DT40-WT, DT40-PKCβ-KO, and DT40-PKCβ-Rescue B cells. (**K**) Statistical comparison of the MFI of pFAK molecules within the B cell immunological synapse of DT40-WT, DT40-PKCβ-KO, and DT40-PKCβ-Rescue B cells in response to antigen stimulation. In (**A**)–(**F**) and (**H**)–(**K**), bar represents mean ± SEM from at least 30 cells in one representative of three independent experiments. Two-tailed *t* tests were performed for statistical comparisons.

The following figure supplement is available for figure 6:

**Figure supplement 1.** PKCβ-dependent FAK activation accounts for B cells to discriminate substrate stiffness.

## PKCβ-dependent FAK activation accounts for B cell discrimination capability by potentiating B cell spreading and adhesion responses

We next investigated how PKCβ-dependent FAK activation accounts for the capability of B cells to discriminate the stiffness of antigen-presenting substrates. We hypothesized that stronger FAK activation in B cells that were placed on stiff substrates leads to better B cell spreading and adhesion responses, both of which are known to be essential for efficient B cell activation. To assess this, we first performed correlation analyses of the pFAK MFI value with each of the following three parameters in the same B cell: (1) the size of the B cell contact interface (or the size of the B cell IS); (2) the strength of B cell adhesion, and (3) the BCR MFI value. Quantifications of the size of the B cell contact interface and the BCR MFI within the contact area were introduced above. The strength of B cell adhesion was quantified by interference reflection microscopy (IRM) following our published protocol (*Xu et al., 2015*). As shown in the representative image in *Figure 7A*, we quantified the MFI of IRM images by subcellular level analysis through ImageJ Software; the darker the region in the IRM image, the smaller the MFI of IRM, and thus the stronger the B cell adhesion, and vice versa (*Figure 7A*). We found that the pFAK MFI value was strongly correlated with each of these three parameters (*Figure 7B–D*). As B cells that were placed on stiff substrates always displayed a much higher pFAK MFI value, as mentioned above, it was not unexpected that these B cells also presented a much higher BCR MFI value and thus a higher BCR MFI ratio index than B cells that were placed on soft substrates (*Figure 7E*). These correlation analyses demonstrated that the enhanced FAK activation in B cells on stiff substrates leads to better B cell spreading and adhesion toward the antigen-presenting surfaces. This suggests that PKCβ-dependent FAK activation accounts for the B cell discrimination capability by potentiating B cell spreading and adhesion responses.

## Adhesion molecules greatly enhanced the capability of B cells to discriminate substrate stiffness

In the previous experiments, although only antigens and not adhesion molecules were present on the substrate surfaces, B cells were still able to detect substrate stiffness during their activation. This suggests that this capability of B cells is independent of the interaction between adhesion and integrin molecules. To test this directly, we used EDTA to block the function of integrins that may be activated by the presence of ECM molecules in the serum and/or the blocking reagent in the incubation buffer. EDTA-treated B cells lost the capability to discriminate substrate stiffness (*Figure 8A*). Although EDTA can interfere with the function of integrin, it is also a potent calcium chelator with a much broader impact on the activation and function of B cells, including BCR-antigen binding-mediated calcium influx. Thus, to confirm this conclusion we used splenic primary B cells from CD11a (ITGAL) knockout (KO) mice, which lack the lymphocyte function-associated antigen 1 (LFA-1) (*Figure 8B*). As the adhesion and activation of lymphocytes requires integrin and adhesion molecules in vivo under physiological conditions (*Arana et al., 2008a*), it could follow that the capability of B cells to discern substrate stiffness is also regulated by direct activation of integrins with external adhesion molecules. B cells mainly express two types of integrins, LFA-1 and VLA-4, that bind to their respective adhesion molecules, ICAM-1 and VCAM-1 (*Arana et al., 2008a*). Thus, we compared

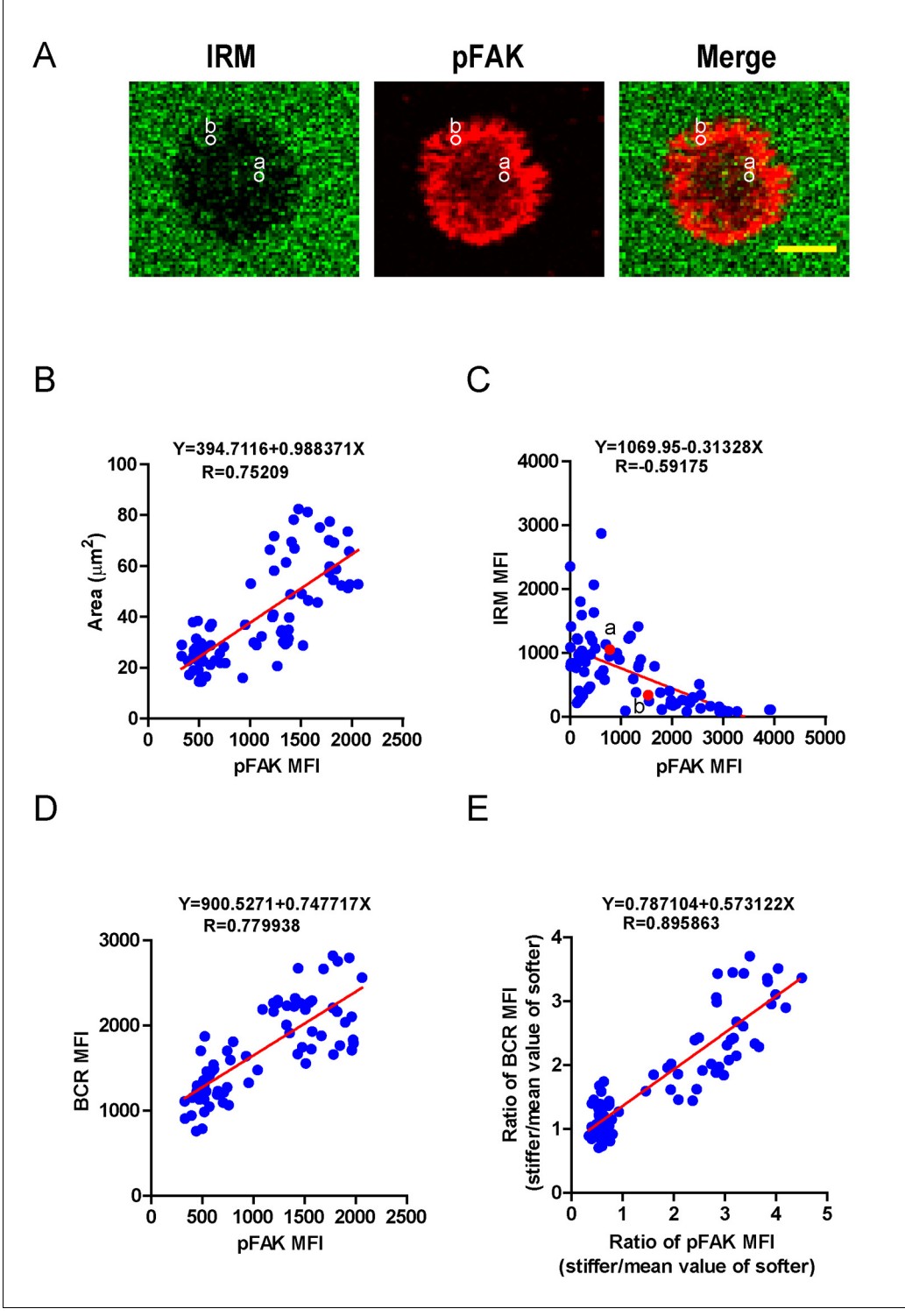

**Figure 7.** PKC$\beta$-dependent FAK activation accounts for B cell discrimination capability by potentiating B cell spreading and adhesion responses. (**A**) Representative confocal images showing the adhesion strength of B cells on the basis of IRM. In both IRM and pFAK images, two representative region of interests (ROIs, (**a and b**) demonstrated the calculation of the IRM and pFAK MFI within the same ROI. Scale bar is 4 µm. (**B–D**) Correlation analysis of the pFAK MFI with the size of spreading area (**B**), the adhesion strength on the basis of IRM MFI (**C**), or the BCR MFI (**D**). (**E**) Correlation analysis of the ratio of the pFAK MFI to the ratio of BCR MFI. Data in B, D, E were analyzed based on the contact area of a single cell, thus one dot represents one cell; while data in (**C**) were analyzed based in a ROI within a cell's contact area as demonstrated in (**A**), thus one dot represents one ROI. In

*Figure 7 continued on next page*

*Figure 7 continued*

(B)–(E), inserted correlation function was the linear regression analysis; data are one representative of at least two independent experiments.

in parallel how NP-specific B1-8 primary B cells discriminate between stiff and soft PDMS substrates that were coated with antigens alone or coated with both antigens and adhesion molecules. The results showed that the addition of either ICAM-1 or VCAM-1 significantly enhanced the B cell's capacity to discriminate between stiff and soft substrates (*Figure 8C–D,G*). Similar results were also acquired using mouse CH27 B cells (*Figure 8E–G*). Thus, during the initiation of B cell activation, integrin signaling plays an important role in maintaining the substrate stiffness discrimination capability of B cells, which is greatly enhanced by the outside-in activation of integrin by adhesion molecules.

## RA patient B cells exhibit a disordered and weakened capability to discriminate substrate stiffness

Lastly, we examined the physiological and pathological relevance of our finding that PKC$\beta$-dependent FAK activation accounts for B cell discrimination of substrate stiffness. A very recent study showed that auto-reactive human primary B cells from RA patients can efficiently acquire the auto-antigen, aggrecan, in a BCR- and adhesion-dependent manner (*Ciechomska et al., 2014*). Thus, we compared the activation of primary B cells placed on either stiff or soft PDMS substrates from either healthy controls or RA patients. To reduce inter-sample and inter-batch variations, we chose nine age- and gender-matched pairs of healthy controls and RA patients. In each batch, we only compared one pair of samples, a healthy individual versus an RA patient. We pre-labeled PBMC B cells from the paired samples using Alexa Fluor 647-conjugated Fab fragment anti-human IgM constant region antibodies and placed these cells on either stiff or soft PDMS substrates presenting anti-human Ig$\kappa$ and anti-human Ig$\lambda$ antibodies, which functioned as the surrogate antigens. The cells were in contact with the antigen-coated PDMS substrates for 15 min before image acquisition. The accumulation of BCR (or pFAK) at the contact site between the B cell and the antigen-presenting substrates was quantified, and the BCR (or pFAK) MFI ratio index values were calculated as above. The results showed that all the BCR (or pFAK) MFI ratio index values were larger than 1, indicating that all the human primary B cells from both healthy controls and RA patients exhibited substrate discrimination capabilities (*Figure 9A–D*). However, careful examination of the BCR MFI ratio index values of the paired samples in the same batch of experiments revealed the presence of different efficiencies in terms of the capability of B cells to discriminate substrate stiffness: B cells from RA patients exhibited a much weaker discrimination efficiency as indicated by the much lower BCR (or pFAK) MFI ratio index, in comparison with B cells from the paired healthy controls (*Figure 9A,B*). We observed this phenomenon in six out of the nine paired samples that we examined (*Figure 9—figure supplement 1A–F*). In two pairs, we observed a comparable B cell discrimination capability in both the healthy control and RA patient (*Figure 9—figure supplement 1G–H*). In only one pair, B cells from the healthy control showed weaker discrimination capability than those from the RA patient (*Figure 9—figure supplement 1I*). It should be noted that the pFAK MFI ratio index value was more in line with this conclusion than the BCR index value, as indicated by much larger differences when comparing the ratio index of healthy controls versus RA patients within the paired samples (*Figure 9C,D*).

Furthermore, we examined whether alteration in the ability of RA patient B cells to discriminate substrate stiffness of the antigen-presenting surface resulted from changes in BCR MFI (as a parameter indicating BCR microclustering) on a stiff or a soft substrate surface. We did so by comparing the BCR MFI of B cells from the paired healthy controls versus the RA patients on either stiff or soft substrates (*Figure 9—figure supplement 2A–I*). The results showed that B cells from healthy controls preferentially enhanced the synaptic accumulation of BCR microclusters on stiff substrates. In direct contrast, RA patient B cells preferentially enhanced BCR accumulation on soft substrates (*Figure 9—figure supplement 2A–I*). We also examined the original MFI value of pFAK and obtained similar results (*Figure 9—figure supplement 3A–I*).

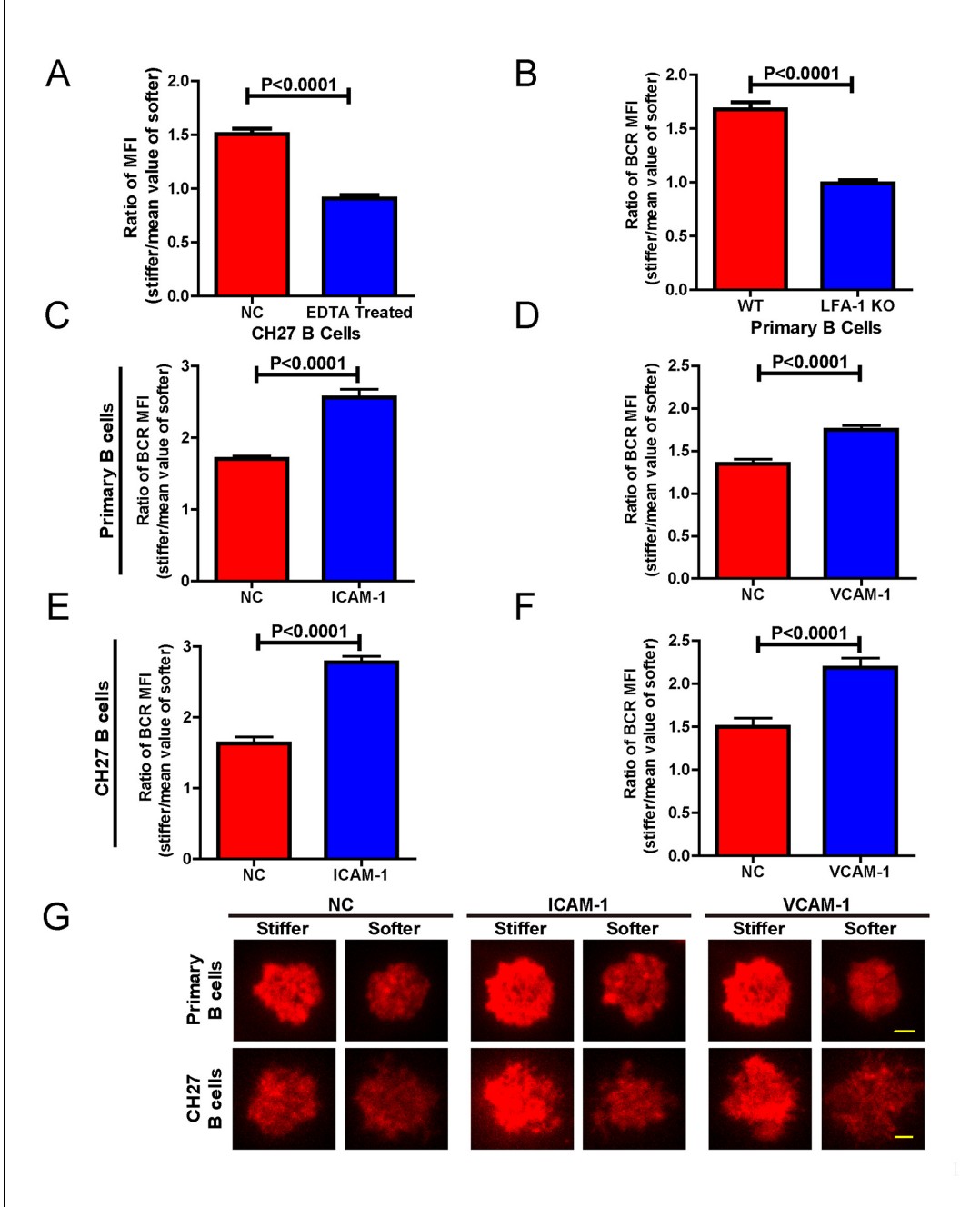

**Figure 8.** Adhesion molecules enhance B cell's capability to discriminate between stiff and soft substrates. (**A**) Blocking the integrin with EDTA reduces the ratio of BCR MFI of CH27 B cells. (**B**) LFA-1 KO primary B cells lost the substrate stiffness discrimination compared with the WT B cells. (**C–F**) Adhesion molecules, ICAM-1 and VCAM-1, enhanced the B cell's capability to discriminate between stiff and soft substrates as shown in B1-8 primary B cells (**C, D**) or CH27 B cells (**E, F**). (**G**) Representative confocal images showing the synaptic accumulation of BCRs from either B1-8 Primary B cells or CH27 B cells that were placed on antigen-presenting substrates with the additional condition of lacking (NC) or presenting adhesion molecules. Scale bar is 3 µm. Bar represents mean ± SEM from one representative of three independent experiments. Data were from at least 20 cells. Two-tailed *t* tests were performed for statistical comparisons.

All of these outcomes further support our model that PKCβ-mediated FAK activation accounts for B cell discrimination of substrate stiffness. In conclusion, the capability of B cells to discriminate substrate stiffness features can be readily recapitulated in human primary PBMC B cells, and more importantly, RA patient B cells exhibited a disordered capability of discriminating substrate stiffness

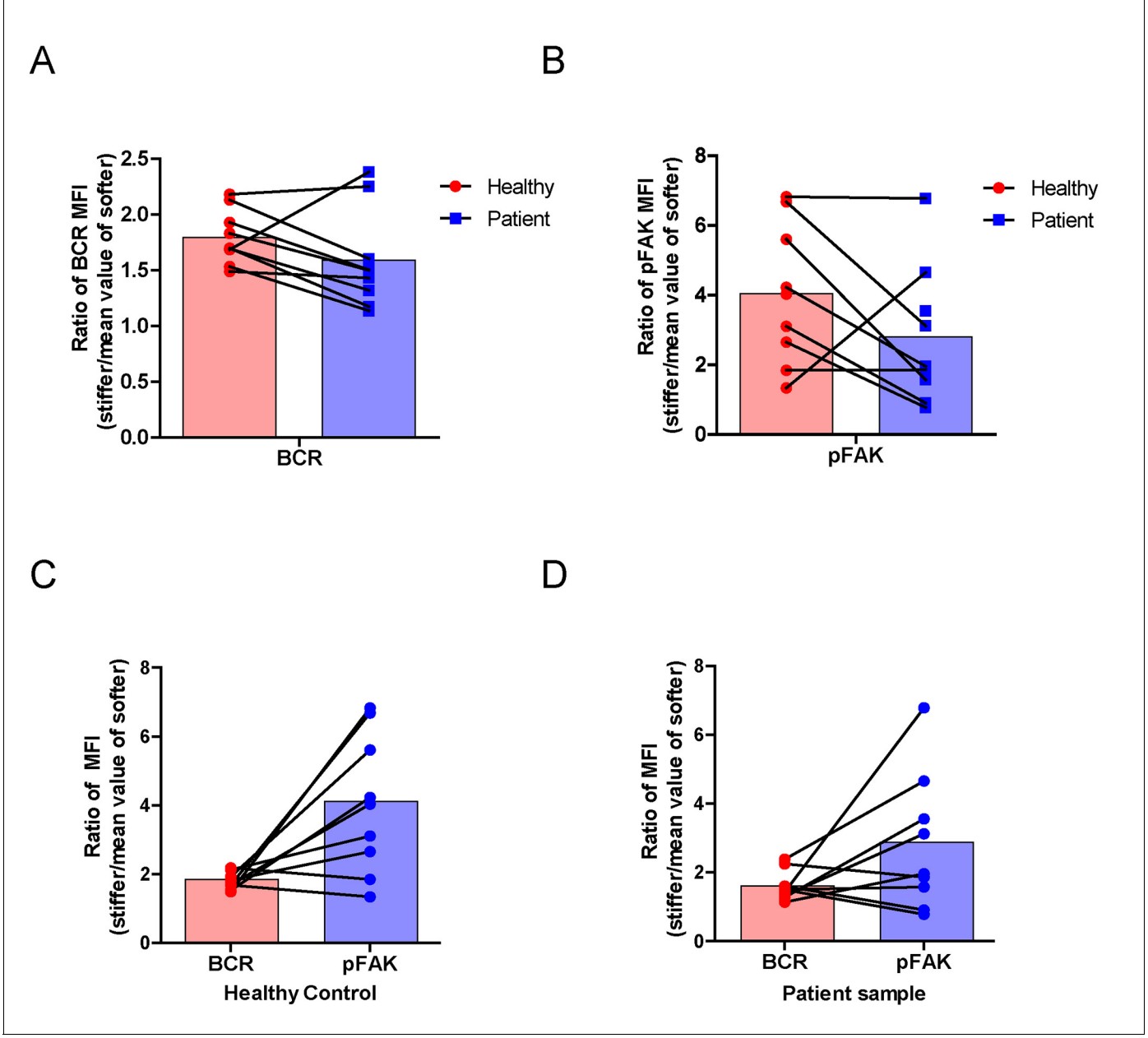

**Figure 9.** RA patient B cells exhibited disordered capability to discriminate substrate stiffness. (A, B) Paired comparison of healthy control and RA patient B cells on the basis of ratio of either BCR MFI (A) or pFAK MFI (B). (C, D) Paired comparison of the ratio of BCR MFI and pFAK MFI on the basis of either healthy control or RA patient B cells.

The following figure supplements are available for figure 9:

**Figure supplement 1.** RA patient B cells exhibited disordered capability to discriminate substrate stiffness.

**Figure supplement 2.** RA patient B cells exhibited disordered capability to discriminate substrate stiffness.

**Figure supplement 3.** RA patient B cells exhibited disordered capability to discriminate substrate stiffness.

in comparison with healthy controls. We discuss how these findings may help explain the dysregulated activation of auto-reactive B cells in RA patients (see Discussion section below).

## Discussion

Initiation of B cell activation has been shown to be sensitive to antigen density (*Liu et al., 2010a*; *Fleire et al., 2006*), antigen affinity (*Liu et al., 2010a*; *Fleire et al., 2006*), antigen valency (*Bachmann et al., 1993*; *Liu and Chen, 2005*; *Liu et al., 2004*), Brownian mobility of antigen (*Wan and Liu, 2012*), and the stiffness feature of the substrates tethering the antigens (*Wan et al., 2013*; *Zeng et al., 2015*). Remarkably, this suggests that B cells use mechanosensing capability to sense the biochemical and biophysical properties of the antigens and their presenting substrates. Recent studies suggested that antigens encountered by B cells *in vivo* are actually presented on substrates with diverse stiffness (*Bachmann and Jennings, 2010*). The stiffness feature of the substrates presenting the antigen can greatly influence the initiation of B cell activation by regulating the efficiency of the accumulation of BCR and antigen molecule into the B cell immunological synapse (*Wan et al., 2013*; *Zeng et al., 2015*; *Wan et al., 2015*).

Here we investigated the underlying molecular mechanisms used by B cells to discriminate substrate stiffness. We found that synaptic recruitment of BCRs is significantly enhanced on activation by antigens on stiff substrates compared with antigens on soft substrates. Our experimental system was very similar to that used by Kam and colleagues, in which only IgG anti-CD3 and anti-CD28 surrogate antigens were tethered to PA substrates (*Judokusumo et al., 2012*), and to the system used by Milone and colleagues in which only IgG anti-CD3 and anti-CD28 surrogate antigens were tethered to PDMS substrates (*O'Connor et al., 2012*). In these systems, no adhesive ligands were used, but, in both cases, T cells similarly showed strong mechanosensing abilities. These T cell studies, in addition to the B cell studies presented in this report, show that mechanosensing by lymphocyte cells may not solely function through direct interaction between ICAM-1 and the well-characterized mechanosensor LFA-1 (*Chen et al., 2010*, *2012*). However, direct ICAM-1 and LFA-1 interactions may still be required to maintain and regulate the mechanosensing ability of lymphocyte cells as other studies have shown that they can fine-tune the activation of both B and T cells (*Carrasco et al., 2004*; *Arana et al., 2008a*; *McLeod et al., 2004*; *Spaargaren et al., 2003*; *Arana et al., 2008b*). Considering this, in this study, we show that the presence of adhesion molecules (ICAM-1 or VCAM-1) greatly enhanced the B cells' capability to discriminate between the degrees of stiffness. Interestingly, recent studies indicate that germinal center B cells (GCBs) recognize antigens on antigen-presenting cells through a specialized immune synapse architecture that is distinct from that of mature naïve B cells (*Nowosad et al., 2016*). Moreover, it has also been reported that specialized antigen-presenting cells, follicular dendritic cells, are usually stiff cells that can promote efficient antigen extraction and stringent affinity discrimination of GCBs, while regular dendritic cells that are mainly responsible for the antigen presentation for mature naïve B cells are mostly soft cells (*Spillane and Tolar, 2017*). Based on these published studies and the data in this report, we propose that adhesion molecules on antigen-presenting cells in germinal centers play an important role in enhancing the activation of GCBs. Indeed, this is supported by a recent study showing that integrin-–ligand interaction within the germinal center B cells and the antigen-presenting FDCs are important for the responses in GC (*Wang et al., 2014*).

In this report, we investigated the molecular nature of the mechanosensor machinery used by B cells to distinguish substrate stiffness in the absence of adhesion molecule-triggered integrin activation. We used a library of chicken DT40 B cell lines deficient for specific signaling molecules (*Kurosaki et al., 2010*; *Kurosaki, 1999*), including Lyn, Syk, PLCγ2, Btk, BLNK, PKCβ, and Carma-1, to dissect the underlining molecular mechanism. The first striking observation was that only BCR signaling-dependent, not BCR signaling-independent, accumulation of BCRs into the B cell IS is subjected to strict regulation by the mechanosensing capability of B cells. Published studies and the data in this report show that BCR signaling-independent accumulation of BCRs and/or antigens was likely induced by the passive trapping of BCRs and antigens into the B cell IS, as both types of molecules exhibit free Brownian thermal diffusion before recognition (*Weber et al., 2008*; *Liu et al., 2010a*, *2010b*; *Tolar et al., 2009*). These published studies also indicate that both signaling-independent and signaling-dependent mechanisms account for accumulation of BCRs into the BCR microclusters. These two steps are not mutually exclusive, but are instead closely related in a

sequential and synergistic way to maximize efficient BCR clustering. Specifically, signaling-independent BCR clustering initiates the earliest signaling on antigen and BCR recognition, which further enhances BCR clustering in a signaling-dependent manner. Our data in this report show that the initial signaling-independent accumulation of BCRs into the BCR microclusters is not sensitive to the stiffness features of the substrates, whereas the subsequent signaling-dependent accumulation of BCRs into the BCR microclusters is. Based on all these studies, it is reasonable to conclude that thermal diffusion-mediated and BCR signaling-independent passive trapping of BCR molecules is not sensitive to substrate stiffness as the Brownian diffusion of BCRs on the plasma membrane would not be affected by the stiffness of the substrates.

In this report, major efforts were made to address how BCR signaling transduction enables B cells to discriminate substrate stiffness during initiation of B cell activation. Mechanistically, the results indicate that PKC$\beta$ functions downstream of Btk and PLC$\gamma$2, as PMA-induced activation of PKC$\beta$ can bypass the requirements of Btk and PLC$\gamma$2 for discriminating substrate stiffness, demonstrating the vital importance of PKC$\beta$ in mediating the capability of B cells to discriminate substrate stiffness. We excluded an involvement for PKC$\beta$-mediated NF-$\kappa$B activation by showing that Carma1-KO B cells maintained their discrimination capability. This result makes sense as the activation of NF-$\kappa$B leads to alterations in gene expression profiles in B cells at a later time point (on the scale of hours), while the substrate stiffness discrimination capability of B cells kicks in minutes after the initial BCR and antigen interaction. Then, the next question becomes which molecular mechanisms determine this dependence of PKC$\beta$ and other upstream BCR signaling molecules for the substrate stiffness discrimination of B cells? Some interesting insight came from a study showing that membrane-proximal BCR signaling molecules including Btk, PLC$\gamma$2, BLNK, and PKC$\beta$ are essential for the inside-out activation of integrin molecules (*Arana et al., 2008a*, *2008b*; *Spaargaren et al., 2003*). Taking this into account with our findings that an outside-in activating signal of integrin significantly enhanced the capability of B cells to discriminate substrate stiffness, we validated our hypothesis that PKC$\beta$-dependent FAK activation accounts for the substrate stiffness discrimination capability of B cells. FAK plays a key role in the activation of integrin signaling pathways by providing a key docking site for Src family kinases, which in turn phosphorylate downstream signaling molecules (*Yu et al., 2012*; *Bashour et al., 2014*; *Slack-Davis et al., 2007*). Specifically, early biochemical studies demonstrated that FAK phosphorylation at Tyr-925 regulates cross-talk between focal adhesion turnover and cell protrusion in embryonic fibroblasts (*Deramaudt et al., 2011*). Indeed, FAK inactivation completely blunted the capability of the B cells to discriminate substrate stiffness, which could be rescued by the exogenous expression of FAK-WT but not the inactivated mutant at Tyr-925. Mechanistically, we found that FAK activation is triggered by BCR engagement in a PKC$\beta$-dependent manner. These findings are consistent with earlier studies showing that integrin-induced FAK phosphorylation can be blocked by inhibiting PKC, and that the PKC activator, PMA, drastically enhanced phosphorylation, and thus activation, of FAK (*Disatnik and Rando, 1999*). Further colocalization analyses in this report indicated that pFAK mainly localized to the peripheral region of the B cell IS, and highly colocalized with F-actin, strongly suggesting that FAK activation supported B cell spreading and adhesion responses (*Figure 9—figure supplement 1J*). Indeed, additional analyses demonstrated that PKC$\beta$-dependent FAK activation accounts for B cell discrimination of stiffness by potentiating B cell spreading and adhesion responses. These findings in B cells are consistent with an earlier study showing that the spreading and adhesion responses of muscle cells are regulated by typical PKC-mediated FAK activation (*Disatnik and Rando, 1999*). A similar observation was also reported in the case of snail defense cells from Lymnaea stagnalis (*Disatnik and Rando, 1999*; *Walker et al., 2010*).

What is the physiological significance of the core finding in this report that PKC$\beta$-dependent FAK activation accounts for B cell discrimination against substrate stiffness? We propose two obvious applications:

Firstly, we propose that B cells have evolved to maintain a high efficiency to discriminate substrate stiffness features because non-self-antigens presented by viral capsids usually exhibit a high degree of stiffness (45,000–1,000,000 kPa) (*Mateu, 2012*). In contrast, self-antigens presented by the plasma membrane usually show a low level of stiffness (0.01–1000 kPa) (*Nemir and West, 2010*) and soluble self-antigens in humoral microenvironment usually display a particularly low degree of stiffness (several Pa) (*Araujo et al., 2012*). We believe that our findings here provide a new explanation for the widely accepted observations in vaccine research and administration that viral like

particle (VLP) antigens are more potent than soluble antigens to induce antibody responses (*Bachmann and Jennings, 2010*; *Bachmann et al., 1997*, *1993*).

Secondly, this report also suggests that RA patient B cells exhibit a disordered capability to discriminate substrate stiffness. B cells from RA patients exhibited weaker discrimination capability than B cells from healthy controls. Specifically, B cells from healthy controls preferentially enhanced the synaptic accumulation of BCR microclusters on stiff substrates. In marked contrast, B cells from RA patients exhibited a different preference of enhancing the accumulation on soft substrates. As B cells from RA patients are known to exhibit hyper BCR signaling (*Nakken et al., 2011b*; *Oligino and Dalrymple, 2003*; *Szodoray et al., 2006*), it is our speculation that the enhanced activation of RA patient B cells on soft substrates results from signaling-dependent accumulation of BCRs into the BCR microclusters, as discussed above. These findings have obvious clinical relevance as there is a diverse range in stiffness features of the antigen-presenting substrates as well as changes in substrate stiffness at the physiological level versus pathological conditions which are associated with disease (*Knight, 2015*). It is reported that auto-reactive human primary B cells from RA patients can efficiently acquire the auto-antigen aggrecan, in a BCR and adhesion-dependent manner (*Ciechomska et al., 2014*) and RA patients usually exhibit highly enriched B cells in the synovia (*Nakken et al., 2011a*). It is expected that the altered stiffness properties of the ECM, which displays the auto-antigens, can drive the auto-reactive B cells to break the anergy state and instead undergo aberrant activation. Indeed, our finding is well supported by a report showing that reduced cartilage stiffness renders B cells into auto-antigen presenters in RA patients, which subsequently causes production of auto-antibodies (*Mauri and Ehrenstein, 2007*). As the ECM-associated microenvironment provides an abundant source of antigens, it has been proposed that any change in ECM stiffness could be identified as a threatening signal by the immune system and thus trigger a response from immune cells (*Tesniere et al., 2008*; *Schaefer, 2010*; *Knight, 2015*). Taking our findings into consideration, it is an intriguing hypothesis that the capability of B cells to sense the stiffness features of antigen-presenting surfaces may be potentially related to detection of that danger by the immune system (*Knight, 2015*).

Conclusively, all these data shed light on the precise molecular mechanism of how B cells discriminate substrate stiffness in a PKC$\beta$- and FAK-dependent manner during initiation of B cell activation, improving our understanding of the sophisticated mechanosensing capability of B cells. We also propose that the mechanosensing and mechanotransducing abilities of immune cells deserve further investigation as these studies could enhance our understanding of immune cell activation on antigen recognition and may help build better vaccines to ultimately cure autoimmune diseases.

## Materials and methods

### B cells, antigens and antibodies

CH27 B cell line (RRID:CVCL_7178, Source: mouse lymphoma) was gifted by Dr. Susan K. Pierce (NIAID-NIH) that was originally purchased from ATCC (USA). Similarly, B1-8 specific primary B cells were negatively selected from IgH B1-8/B1-8 Igκ $-/-$ transgenic mice as described previously (*Liu et al., 2010a*). All the chicken DT40 B cells, including DT40-WT (RRID:CVCL_J437), DT40-LYN KO (RRID:CVCL_1T41), DT40-SYK KO (RRID:CVCL_1T43), DT40-PLCγ2 (RRID:CVCL_1T47), DT40-BTK (RRID:CVCL_1T45), DT40-BLNK KO (RRID:CVCL_1T38) and DT40-PKC$\beta$ KO (RRID:CVCL_1T42) were gifts for laboratory scientific studies from Dr. Tomohiro Kurosaki (RIKEN, Japan). CD11a (ITGAL) KO mice were gifted by Dr. Yan Shi (Tsinghua University, China). Mice B cell lines including CH27 and mouse primary naïve B cells were cultured in RPMI-1640 medium supplemented with 10% FBS, 50 μM $\beta$-mercaptoethanol (Sigma-Aldrich), and penicillin/streptomycin antibiotics (Invitrogen). DT40 chicken B cell lines including WT and KO used in this study were maintained at 37°C in the same medium as above, but 1% chicken serum was added into the above mentioned culture medium. 293 T cells were purchased from Cell bank (Chinese academy of sciences, Shanghai). 293 T cells were maintained in the DMEM culture medium supplemented with 10% FBS, and penicillin/streptomycin antibiotics (Invitrogen). All cell lines used in this study were negative for mycoplasma contamination test using a PCR detection method.

Mouse anti-chicken IgM antibody (clone M1 or M4) (cat# 8310–01 or cat# 8300–08) and Goat anti-mouse Igκ kappa light chain, Goat anti-human kappa and lambda light chain antibodies were

purchased from Southern Biotech. DyLight 649-conjugated Fab anti-mouse IgM constant region antibodies, Alexa Fluor 647 conjugated Fab fragment of IgM constant region antibodies anti-human, Goat F(ab)$_2$ anti-mouse IgM + IgG (H+L) (Lot# 119316) and Cy5-conjugated Fab Goat anti-mouse IgM were purchased from Jackson ImmunoResearch. Phospho-FAK (Try 925) (RRID: AB_10831810) was purchased from Cell Signaling. Labeling mouse anti-chicken IgM antibody (clone M1) with Alexa Fluor-647 and digesting the Fab fragment of mouse anti-chicken IgM antibody (clone M1) were done following our published protocol (Liu et al., 2010b, 2010a). NP8-BSA was obtained from Bioresearch technology. Rabbit polyclonal anti-mouse FAK antibodies (Cat# AMO0672) were purchased from ThermoFisher.

## Plasmid constructs

Chicken Lyn, Syk and PLCγ2, Btk and PKCβ were cloned from DT-40 cDNA and constructed by fusing GFP with the C-terminal, then incorporated into pEGFP vectors. Chicken FAK was cloned from DT40-WT cDNA and constructed by fusing mCherry with the N-terminal, then incorporated into pHAGE vector. A PCR-based mutagenesis strategy was used to construct pHAGE-mCherry-FAK-Y926F plasmid through Gibson Assembly. pSpCas9-2a-GFP was a gift from Dr. Feng Zhang (MIT, Cambridge).

## Generation of gene FAK KO cell line

FAK was knocked out in DT40 B cells using the CRISPR/Cas9 technique. Guide RNA was designed using the website http://crispr.mit.edu. Two target sites on the N-terminal and C-terminal of the gene were used to promote the KO efficiency, and the sequences of those are AACCTTTAGGAC TCGCTCCA and GGCTGGTCATGACGTACTGC. The DT40 B cells transiently expressing pSpCas9-2a-GFP by electroporation were sorted and cultured in a 96-well plate. The KO cells were detected by PCR and western-blot.

## Transfection and retroviral transduction

For DT40 B cell electroporation, the Buffer T and B-009 program of Amaxa Nucleofector was used according to the protocol from Lonza. For the retroviral transduction, 293 T cells were transfected with pHAGE and packaging plasmids by the calcium phosphate method. After 48 hr at 37°C ecotropic viral supernatants were collected and added to B cell culture medium in the presence of 5 μg/mL polybrene. Positive cells were sorted by flow cytometry. Cell sorting was performed using the FACSAria III Cell Sorter (BD), by following the BD protocols.

## Western blotting

2×Lysis buffer containing Tris-Hcl pH 7.4 50 mM, NaCl 150 mM, EDTA 20 mM and NP40 4% was used to lyse 2 × 10$^6$ DT40 WT and FAK-KO B cells. Protein was separated by 10% Bis-Tris PAGE (Life technologies), and then transferred to a polyvinylidene fluoride (PVDF) membrane. FAK was probed with primary antibody, Rabbit pAb anti-mouse FAK (RRID:AB_1500093), and then appropriate HRP-conjugated secondary antibodies (Dako).

## Integrin blocking assay

We used 5 mM concentration of EDTA in PBS containing 1% FBS. We pre-treated 2 × 10$^6$ CH27 B cells with EDTA-containing buffer for 20 min at 37°C, washed the cells twice with 1X PBS and then proceeded to the next step.

## RA patients and healthy control subjects

A total of nine RA patients, matching the criteria of RA disease according to the American College of Rheumatology, were enrolled. For the control group healthy volunteers were recruited. This study was approved by the committee of ethics at Beijing People's Hospital of Peking University. Each healthy volunteer and patient submitted their informed consent. A total of 8 ml of peripheral blood was acquired from each person. PBMC were isolated from the healthy and RA patient samples using Ficoll-Paque plus density separation, and were frozen at −80 under liquid nitrogen prior to use. Before use for imaging experiments, human PBMS cells were cultured in a RPMI-1640 medium supplemented with 10% FBS, 50 μM β-mercaptoethanol (Sigma-Aldrich), and penicillin/streptomycin

antibiotics (Invitrogen) for 2–3 hr. Cells were stimulated with Goat anti-human kappa and lambda light chain antibodies. For staining the BCRs of PBMS, Alexa 647 conjugated Fab fragment of IgM constant region anti-human antibodies were used.

## Preparation of polyacrylamide (PA) substrates to present antigens

To prepare antigen-presenting polyacrylamide gel (PA) substrates, glass coverslips were treated with NaOH, 3-aminopropyltrimethoxysilane, and glutaraldehyde in a step-by-step manner. After washing extensively in distilled $H_2O$, the glass coverslips were ready to support the polyacrylamide gels. The rigidity of the polyacrylamide gel substrate was controlled using different amounts of bisacrylamide cross-linker while keeping the total acrylamide concentration constant at 10% (w/v). In our report, polyacrylamide gels with different Young's modulus were produced using bisacrylamide concentrations of 0.8% and 0.05% (w/v). Antigens were tethered to the surface of polyacrylamide gel substrates following methods described previously (*Judokusumo et al., 2012*). In our report, streptavidin-conjugated acrylamide was polymerized into the polyacrylamide gel for the purpose of tethering the biotinylated F(ab')two anti-IgM antibody as surrogate antigen, which was generally incubated with the substrate at 37°C for 30 min at a concentration of 30 µg/ml. After extensive washing, the polyacrylamide gel substrate tethering specific antigens were ready for use (*Wan et al., 2013*).

## Preparation of poly (dimethylsiloxane) substrates to present antigens

Poly-dimethylsiloxane (PDMS) substrates were prepared following a standard protocol from our previously published studies (*Zeng et al., 2015*). In brief, we prepared the PDMS substrates by mixing dimethylsiloxane monomer (Dow Corning Sylgard 184) with a cross-linking agent, according to the manufacturer's instructions. The ratio of cross-linking agent to base polymer was 1:5 (Stiff) or 1:50 (Soft) to prepare PDMS substrates with different stiffness features. After defoaming in a mixer and degassing under vacuum, PDMS elastomers were cured at 60°C for 4 hr on glass coverslips or in 24-well-cell culture plates. PDMS forms a planar surface with highly hydrophobic features that can easily tether proteins through adsorption. To coat antigens which can activate B cells, PDMS substrates were coated with 5 µg/ml NP8-BSA or 5 µg/ml anti-BCR surrogate antigens in PBS as surrogate antigen overnight at 4°C. After washing with PBS, the PDMS elastomers were blocked with 5% BSA in PBS at 37°C for 1 hr, followed by thorough washing before downstream experiments (*Zeng et al., 2015*)

## Evaluation of the antigen density on the surface of PDMS elastomer substrates

Alexa Fluor 647-conjugated mouse anti-chicken IgM antibody (clone M4) was used on PDMS surfaces to examine the tethered concentration of antigen, then imaged by confocal fluorescence microscopy (Zeiss, LSM710). To determine the antigen accessibility towards antibody, mouse anti-chicken IgM antibody (clone M4) was incubated on soft and stiff PDMS substrate as surrogate antigen overnight at 4°C. After washing with PBS, it was blocked with 5% BSA for 1 hr at 37°C and then washed thoroughly. 100 nM Alexa DyLight 649-conjugated Fab anti-mouse IgM constant region antibody was added and incubated for 1 hr at 37°C, and then images were acquired after washing thoroughly. The mean fluorescence intensity (MFI) was analyzed by Image J (NIH, U.S.). For cell accessibility, DT40 B cells were loaded on both the surfaces for 10 min at 37°C. Adhered B cells were imaged under a microscope before washing or after washing with 10 ml PBS-1% FBS at the speed of 0.5 or 1 ml/s. Images were acquired and quantified for the number of B cells adhered before washing and the number left on the PDMS surface after washing.

The rate of adhesion was quantified according to the following equation:

*Adhesion Rate = the number of cells in a defined area on the gel surface after washing/the number of cells on the same area of the gel surface before washing*

## Use of adhesion molecule ligands ICAM-1 and VCAM-1 in PDMS system

Recombinant mouse ICAM-1/Fc and VCAM-1/Fc ligands were purchased from Sino Biological Inc and stored at −80°C as per the instructions of the manufacturer. After overnight incubation of surrogate antigen on PDMS at 4°C, ICAM-1 or VCAM-1 (2 µg/ml) were added and incubated for 2 hr at

37°C. After washing, the chambers were blocked with 5% BSA for 30 min at 37°C and washed thoroughly before use. Then, pre-stained cells were loaded into the chamber and incubated for 10 min at 37°C. Cells were fixed with 4% paraformaldehyde at room temperature for 15 min, then washed carefully with PBS and imaged under TIRFM.

## Treatment of B cells with PMA and pharmaceutical inhibitors

Cells were pre-treated with the inhibitors under different conditions following protocols in published studies or manufacturer's instructions as detailed below. Piceatannol (CALBiochem) was used at 50 nM working concentration at RT for 15 min (*Liu et al., 2010a*). PP2 was used at 20 μM (Sigma-Aldrich) (*Liu et al., 2010a*), U73122 at 5 μM (EMD Millipore) (*Mao et al., 2006*), PF573-228 at 1 μM (Sigma-Aldrich) (*Slack-Davis et al., 2007*), and bisindolylemaleimide at 3.5 μM (Sigma-Aldrich) (*Zhou et al., 1999*). DT40 WT and KOs were treated with different concentrations of PMA (5, 20, 50 ,and 100 ng/ml) for 1 hr at RT (Sigma-Aldrich) (*Quann et al., 2009*). After the inhibitor treatment as described above, the cells were washed three times with PBS and were ready for further processing.

## Intracellular staining of signaling molecules

Surrogate antigens were tethered on the PDMS substrates and B cells were labeled with 200 nM of the DyLight 649-conjugated Fab anti-mouse IgM constant region antibodies or Alexa 647 conjugated Fab fragment of anti-human IgM constant region antibodies. After washing twice carefully, cells were stimulated with the surrogate antigens for 15–20 min at 37°C and then were fixed with the 4% paraformaldehyde for 30 min at RT. Cells were washed with the PBS slowly and then treated with the 0.2% Triton-X 100 in PBS for 20 min at RT. Donkey nonspecific IgG (Jackson Immunoresearch Laboratory) was employed for 1 hr at 37°C as a blocking reagent. Phospho-FAK (Try 925) (Cell Signaling) antibody was used as the primary antibody for 1 hr at 37°C. After careful washing, Alexa 555 Donkey anti-rabbit IgG antibody was used as a secondary antibody for 45 min at RT. After careful washing, cells were imaged under a confocal microscope.

## Molecular imaging by TIRFM and confocal fluorescence microscopy

For all of the respective experiments, B cells were obtained from chicken cell line DT40, mice cell line CH27 or mouse primary naïve B cells, or human PBMC cells, and stained with Alexa Fluor 647-conjugated mouse Fab anti-chicken IgM (clone M1), Alexa Fluor 647-conjugated Goat Fab anti-mouse IgM specific for Fc5μ, or Alexa 647 conjugated Fab fragment of IgM constant region antibodies anti-human, respectively. After washing twice they were placed on either the soft or stiff PDMS elastomer surfaces with tethered antigens for 10 or 15 min at 37°C, 5% $CO_2$ depending on the experiment. The cells were then fixed with 4% paraformaldehyde at room temperature for a minimum of 10 min. An Olympus IX-81 Microscope was used to acquire images supported by the Port of TIRF. EMCCD electron-multiplying camera ANDOR iXon + DU897D, 100X Olympus 1.49 NA objective. Lenses with lasers of 488 nm, 561 nm, and 647 nm were used (Sapphire lasers, Coherent). The time of exposure was 100 ms until it was indicated specifically. Metamorph software was used to control the acquisition (MDS Analytical Technologies). The images on PA gel and the IRM images were acquired using scanning laser confocal Olympus FLUOVIEW FV1000 with a 60x oil objective lens. Images were analyzed by Image J (NIH, U.S.) software as our previous studies reported (*Wan and Liu, 2012*; *Liu et al., 2010c*, *2010b*, *2010a*). Mean fluorescence intensity (MFI) value of BCRs, was calculated by Image J software based on mean pixel intensity. Briefly, regions of interest (ROIs) were marked in the images that were already subtracted for background. The MFI values were calculated as the ratio of integrated fluorescence intensity of a ROI to the total area, as our previous studies reported (*Wan and Liu, 2012*; *Liu et al., 2010c*, *2010b*, *2010a*). We calculated the ratio by dividing the BCR MFI of each cell on stiff substrates to the averaged value of the BCR MFI of all the cells on soft substrates. We used that value to quantify the substrate stiffness discrimination capability of B cells. The ratio was obtained from the following equation:

*Ratio = BCRs MFI of each cell on stiff substrates / the averaged value of the BCR MFI of all the cells on soft substrates*

## Acknowledgements

This work is supported by funds from Ministry of Science and Technology of China (2014CB542500 to WLL, 2013CB933702 to CYX) and National Science Foundation China (81422020 and 81621002 to WLL, 11272015 to CYX). We thank Dr. Susan K Pierce (National Institute of Allergy and Infectious Diseases, National Institutes of Health), Dr. Klaus Rajewsky (Immune Regulation and Cancer, Max-Delbrück-Center for Molecular Medicine), Dr. Mark Shlomchik (Yale University), and Dr. Yan Shi (Tsinghua University) for generously providing experimental materials. We also thank Dr. Tomohiro Kurosaki and Dr. Hisaaki Shinohara (WPI Immunology Frontier Research Center, Osaka University, Japan) for generously providing experimental materials.

## Additional information

### Funding

| Funder | Grant reference number | Author |
|---|---|---|
| Ministry of Science and Technology of the People's Republic of China | 2014CB542500 | Wanli Liu |
| National Science Foundation | 81422020 | Wanli Liu |
| National Science Foundation | 81621002 | Wanli Liu |

The funders had no role in study design, data collection and interpretation, or the decision to submit the work for publication.

### Author contributions

SS, ZW, Conceptualization, Data curation, Formal analysis, Validation, Investigation, Methodology, Writing—original draft, Writing—review and editing; ZLi, Conceptualization, Data curation, Formal analysis, Validation, Investigation, Methodology, Writing—original draft, Writing—review and editing, this new author helped in newly done experiments suggested by reviewers especially in FAK Knockout experiment; AC, Resources, Data curation, Formal analysis, Methodology, Writing—review and editing; XL, Data curation, Software, Formal analysis, Methodology, Involved in FAK Knockout and western blot; SZ, Data curation, Software, Formal analysis, Methodology; YLi, JY, YZ, JW, XC, LX, WC, FW, YLu, WZ, Data curation, Software, Formal analysis; YS, Resources, Data curation, Writing—review and editing, assistance in LFA-KO experiments; XS, ZLi, Resources, Software, Formal analysis; CX, Resources, Formal analysis, Visualization, Writing—review and editing; WL, Conceptualization, Resources, Supervision, Funding acquisition, Validation, Visualization, Writing—original draft, Project administration, Writing—review and editing

### Author ORCIDs

Samina Shaheen, http://orcid.org/0000-0003-2890-6827
Xinxin Li, http://orcid.org/0000-0003-0055-1999
Yan Shi, http://orcid.org/0000-0002-6715-7681
Wanli Liu, http://orcid.org/0000-0003-0395-2800

### Ethics

Human subjects: This study was approved by the committee of ethics in Beijing People's Hospital of Peking University. Each healthy volunteer and patient had submitted their informed consent prior to this study.

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
