## [Decision Letter]

Thank you for submitting your article "Substrate stiffness governs the initiation of B cell activation by the concerted signaling of PKCβ and FAK" for consideration by *eLife*. Your article has been favorably evaluated by Tony Hunter (Senior Editor) and three reviewers, one of whom, Michael L Dustin (Reviewer #1), is a member of our Board of Reviewing Editors. The following individual involved in review of your submission has agreed to reveal their identity: Ziv Shulman (Reviewer #2).

The reviewers have discussed the reviews with one another and the Reviewing Editor has drafted this decision to help you prepare a revised submission.

Summary:

The stiffness of antigen-presenting cells play an important role in B cell activation. In the current study the authors solved the connection between mechanical stress and intracellular signals that regulate BCR clustering. They established a system for examination of the role of many signaling molecules in the process by using both genetic and pharmacological approaches. Most impressively, the authors dissected the down-stream signaling components of BCR accumulation in the synapse rather than initial BCR clustering per se. This is a very impressive and innovative approach for studying the process and adds another layer of understanding of the BCR functions.

Essential revisions:

1) The support for a role of FAK is based on a small molecule inhibitor and correlations. Given the initial use of the DT40 cell line, it would be much stronger if they made and tested the FAK knockouts. Then they might also be able to knock in mutants that lack key PKC phosphorylation sites, which would then close a mechanistic gap in the study. With this cell line the knockout is feasible in a couple of months.

2) The authors use BCR clustering as readout; however, how this clustering affects downstream of BCR and cellular activation is missing. There is evidence that cluster size controls B cell signaling but this published data does not incorporate force as a variable. Therefore, direct data on functional effects of these substrates are needed. For instance, how the activation marker CD86 is affected by stimulation on soft or stiff substrates? IgD and CCR7 are also good candidates to examine. Moreover, it would be interesting to show that the stiffness of the substrate affects critical cellular phenotypes such as proliferation (by DAPI or Brdu), differentiation into antibody forming cells (look for CD138+ cells or secreted antibodies) and class switch recombination.

3) There is a problem in description of the PDMS and PA substrates. The authors seem to try to combine the descriptions of the methods for the two substrates, but it is essential that they separate these out and explicitly discuss both substrates individually. As PDMS is a glass like material in many respects, methods like antibody adsorption that work on glass can be applied readily to PDMS. But these same methods don not work with PA gels, which are hydrogels and have very low protein adsorption. Thus, people typically work with PA gels by covalently attaching via attachment sites introduced through crosslinking chemistry. For antibody attachment, most investigators have utilized streptavidin, as the direct covalent methods typically produce too few sites. Thus, the authors need to explain better how the PA gels were functionalized and if they can be compared to the PDMS gels. The results between the two systems are interesting in that they suggest a relationship between substrate stiffness and BCR recruitment may operate over a 2 log range from ~2 kPa to 1000 kPa. The usual reason to use both types of gels is that it has been seen in other systems that the attachment chemistry can strongly influence results such that PA and PDMS substrates behave differently. The observation that both show similar discrimination, although in different ranges, is potentially impressive and convincing, but the attachment of ligands to the PA gel needs to be explained better. This should be straightforward to address, as it is just a matter of better organized Materials and methods section.

4) One primary concern is that the manuscript did not show whether the alterations in the ability of B cells to discriminate high and low stiffness of antigen-presenting surface is due to a decrease in BCR clustering response to antigen on a stiff surface or an increase in response to antigen on a soft surface. Addressing this question is important because inhibition of proximal signaling molecules may primarily block BCR clustering on stiff surfaces while B cells from patients with rheumatoid arthritis may preferentially enhance BCR clustering on soft surfaces. These data are embedded in the ratios of BCR mean fluorescence intensity and can be easily extracted out. Such results potentially provide better explanations for changes in the immunological properties of B cells. This should be a straight forward re-analysis of the existing data.

[Editors' note: further revisions were requested prior to acceptance, as described below.]

Thank you for resubmitting your work entitled "Substrate stiffness governs the initiation of B cell activation by the concerted signaling of PKCβ and focal adhesion kinase" for further consideration at *eLife*. Your revised article has been favorably evaluated by Tony Hunter (Senior Editor), a Reviewing Editor, and two reviewers.

The manuscript has been improved but there are some remaining issues that need to be addressed before acceptance, as outlined below:

For the new data and Discussion provided in the revised manuscript, there two suggestions.

1) The authors should point out that while EDTA can interfere with integrin-mediated cell adherence, as a calcium chelator, EDTA may have a much broader impact on B cells, including BCR-induced calcium influx.

2) The new data that show that B cells discriminate stiff and soft antigen-presenting surface by increasing BCR clustering at the stiff surface, which is dependent on the proximal signaling of the BCR. In contrast, B cells from lupus patients fail to increase BCR clustering on the stiff surface while enhancing BCR clustering on the soft surface. This result should be included in the discussion of signaling-dependent and independent BCR clustering and its implication in B cell activity in lupus patients.

The authors are urged to have the paper read and edited carefully for correct English language usage. We anticipate that a manuscript including these specific changes and with some overall improvement in English editing will be acceptable for publication without a need for re-review. So please do your best in order to minimize delays associated with extensive copyediting.

---

## [Author Response]

*Essential revisions:*

*1) The support for a role of FAK is based on a small molecule inhibitor and correlations. Given the initial use of the DT40 cell line, it would be much stronger if they made and tested the FAK knockouts. Then they might also be able to knock in mutants that lack key PKC phosphorylation sites, which would then close a mechanistic gap in the study. With this cell line the knockout is feasible in a couple of months.*

We have taken this important point into consideration when revising the manuscript. We knocked out FAK in DT40 B cells (DT40-FAK-KO) through CRISPR/Cas9 technique (Figure 6—figure supplement 1 A-C) and found that DT40-FAK-KO B cells completely lost the capabilities to discriminate substrate stiffness during their activation (Figure 6). Furthermore, the exogenous expression of chicken FAK-WT rescued the substrate stiffness discrimination capability during B cell activation (Figure 6). The phosphorylation of FAK at Tyr-925 is considered to be a critical step in FAK activation-mediated migration and adhesion responses (Deramaudt et al., 2011). Not surprisingly, the exogenous expression of chicken FAK-Y926F mutant (chicken FAK carries Tyr 926 that shares sequence homology with Try 925 in mouse and human FAK) (Figure 6—figure supplement 1) in DT40-FAK-KO B cells failed to rescue such capability of B cells (Figure 6). Thus, all these results showed that FAK activation is required for the B cells to discriminate substrate stiffness. We have provided the data and the explanation of these experiments in the Results, Materials and methods and Discussion sections of the revised manuscript.

*2) The authors use BCR clustering as readout; however, how this clustering affects downstream of BCR and cellular activation is missing. There is evidence that cluster size controls B cell signaling but this published data does not incorporate force as a variable. Therefore, direct data on functional effects of these substrates are needed. For instance, how the activation marker CD86 is affected by stimulation on soft or stiff substrates? IgD and CCR7 are also good candidates to examine. Moreover, it would be interesting to show that the stiffness of the substrate affects critical cellular phenotypes such as proliferation (by DAPI or Brdu), differentiation into antibody forming cells (look for CD138+ cells or secreted antibodies) and class switch recombination.*

We fully agree with the reviewer on the importance of these suggested experiments. We wish to explain that the functional effects of substrate stiffness downstream of antigen receptor (both BCR and TCR) and cellular activation have been elucidated in our previous published work for B cells (Wan et al., 2013, J Immunol; Zeng et al., 2015, Euro J Immunol) and in those of others’ published studies for T cells (Judokusumo et al., 2012, Biophys J; O’Connor et al., 2012, J Immunol). In these published studies, the following experiments have been performed to address how substrates stiffness features affect the downstream of BCR and cellular activation: (1) the accumulation of pSyk within the B cell immunological synapse; (2) the upregulation of early activation marker CD69; (3) the proliferation, differentiation and the class switch recombination of activated B cells. Similarly, the downstream of TCR and cellular activation of T cell on stiff versus soft substrates was examined by measuring the following events: (1) the accumulation of pZAP70 and pSFK; (2) IL-2 secretion; (3) polyclonal expansion of peripheral blood T cells; (5) Th1-like differentiation. Thus, these recent studies of ours and those of others showed that the stiffness features of the substrates presenting the antigens affected the downstream of antigen receptor and cellular activation in both B and T cells (Bashour et al., 2014 Proc Natl Acad Sci U S A.; Judokusumo et al., 2012, Biophys J; O’Conner et al., 2012, J Immunol; Wan et al., 2015, *eLife*; Wan et al., 2013, J Immunol; Zeng et al., 2015, Euro J Immunol). However, the underlying molecular mechanism of how substrate stiffness features regulate the initiation of B cell activation remains to be explored. Thus, it would be of great interest to investigate the molecular machinery that is used by B cells and BCRs to discriminate the stiffness features of the substrates presenting antigens during the initiation of B cell activation. In this report, we tried to address this question through a combination of molecular imaging, genetic and pharmacological approaches by examining initiation of B cell activation (as quantified by BCR microclustering and polarization into the B cell immunological synapse) on antigen presenting substrates with either stiff or soft features.

*3) There is a problem in description of the PDMS and PA substrates. The authors seem to try to combine the descriptions of the methods for the two substrates, but it is essential that they separate these out and explicitly discuss both substrates individually. As PDMS is a glass like material in many respects, methods like antibody adsorption that work on glass can be applied readily to PDMS. But these same methods don not work with PA gels, which are hydrogels and have very low protein adsorption. Thus, people typically work with PA gels by covalently attaching via attachment sites introduced through crosslinking chemistry. For antibody attachment, most investigators have utilized streptavidin, as the direct covalent methods typically produce too few sites. Thus, the authors need to explain better how the PA gels were functionalized and if they can be compared to the PDMS gels. The results between the two systems are interesting in that they suggest a relationship between substrate stiffness and BCR recruitment may operate over a 2 log range from ~2 kPa to 1000 kPa. The usual reason to use both types of gels is that it has been seen in other systems that the attachment chemistry can strongly influence results such that PA and PDMS substrates behave differently. The observation that both show similar discrimination, although in different ranges, is potentially impressive and convincing, but the attachment of ligands to the PA gel needs to be explained better. This should be straightforward to address, as it is just a matter of better organized Materials and methods section.*

We agree with the reviewer on this point. To coat antigen molecules to the surface of PA and PDMS substrates, very different coupling procedures shall be applied. In our previously published studies (Wan et al., 2013, J Immunol; Zeng et al., 2015, Euro J Immunol), we have described these methods in detail.Briefly, when preparing antigen presenting polyacrylamide gel (PA) substrates, glass coverslips were treated with NaOH, 3-aminopropyltrimethoxysilane, and glutaraldehyde in a step-by-step manner. After washing extensively in distilled H2O, the glass coverslips were ready to support the polyacrylamide gels. The rigidity of the polyacrylamide gel substrate was controlled by using different amounts of bisacrylamide cross-linker while keeping the total acrylamide concentration constant at 10% (w/v). In our report, polyacrylamide gels with different Young’s modulus were produced by using bisacrylamide concentrations of 0.8 and 0.05% (w/v). Antigens were tethered to the surface of polyacrylamide gel substrates following the methods described previously (Judokusumo et al., 2012, Biophysical J). In our report, streptavidin-conjugated acrylamide was polymerized into the polyacrylamide gel for the purpose of tethering the biotinylated F(ab’)2 anti-IgM antibody as surrogate antigen, which was generally incubated with the substrate at 37℃ for 30 minutes at a concentration of 30 μg/ml as reported in our published studies (Wan et al., 2013, J Immunol). After extensive washing, the polyacrylamide gel substrates tethering specific antigens were ready to use.

Poly-dimethylsiloxane (PDMS) substrates were prepared by following a standard protocol from our published studies (Zeng et al., 2015, Euro J Immunol). In brief, we prepared the PDMS substrates by mixing dimethylsiloxane monomer (Dow Corning Sylgard 184) with cross-linking agent, according to manufacturer instructions. The ratio of cross-linking agent to base polymer was used at 1:5 (Stiff) or 1:50 (Soft) to make PDMS substrate with different stiffness features. After defoaming in a mixer and degassing under vacuum, PDMS elastomers were cured at 60 °C for 4 hours after prepared on glass coverslips or in 24-well-cell culture plates. PDMS forms a planar surface with highly hydrophobic features that can easily tether proteins through adsorption as reported (O’Conner et al., 2012, J Immunol), To coat antigens which can activate B cells, PDMS substrates were coated with 5 µg/ml NP8-BSA or 5 µg/ml anti-BCR surrogate antigens in PBS as surrogate antigen overnight at 4°C. After washing with PBS, the PDMS elastomer were blocked with 5% BSA in PBS at 37 °C for 1 hour and followed by thoroughly washing before downstream experiments (Zeng et al., 2015, Euro J Immunol).

In the Materials and methods section of the revised manuscript, we have provided the detail of coating antigens to the surface of either PA or PDMS substrates respectively.

*4) One primary concern is that the manuscript did not show whether the alterations in the ability of B cells to discriminate high and low stiffness of antigen-presenting surface is due to a decrease in BCR clustering response to antigen on a stiff surface or an increase in response to antigen on a soft surface. Addressing this question is important because inhibition of proximal signaling molecules may primarily block BCR clustering on stiff surfaces while B cells from patients with rheumatoid arthritis may preferentially enhance BCR clustering on soft surfaces. These data are embedded in the ratios of BCR mean fluorescence intensity and can be easily extracted out. Such results potentially provide better explanations for changes in the immunological properties of B cells. This should be a straight forward re-analysis of the existing data.*

We agree with the reviewer on this point. In the revised manuscript, we examined whether the loss of substrate stiffness discrimination ability is due to the changes in the BCR MFI (as a parameter indicating BCR microclustering) on a stiff or soft substrate surface. We achieved this by comparing the BCR MFI of the inhibitor pre-treated versus DMSO-control pre-treated primary B cells (Figure 3—figure supplement 1) on the surface of the substrates with the same Young's modulus in a unit as Pascal (Pa or N/m^2^ or m^−1^·kg·s^−2^). The results readily demonstrated that inhibition of proximal signaling molecules, Lyn, Syk or PLCγ2 primarily blocked the synaptic accumulation of BCRs on stiff substrates, while the changes of BCR MFI on soft substrates were very mild in the comparison of inhibitor pre-treated versus DMSO-control pre-treated primary B cells (Figure 3—figure supplement 1). Similarly, we showed in the revised manuscript that the deficiency of the proximal signaling molecules primarily blocked the synaptic accumulation of BCRs on stiff substrates, while the changes of BCR MFI on soft substrates were just mild in the comparison of KO versus WT B cells (Figure 4—figure supplement 1).

Lastly, we examined whether the alterations in the ability of RA patient B cells to discriminate substrate stiffness of antigen-presenting surface is due to the changes in BCR MFI (as a parameter indicating BCR microclustering) on a stiff or soft substrate surface. We did so by comparing the BCR MFI of B cells from the paired healthy controls versus RA patient on either stiff or soft substrates (Figure 9—figure supplement 2). The results showed that B cells from healthy controls preferentially enhanced the synaptic accumulation of BCR microclusters on stiff substrates, while RA patient B cells exhibited different preference of enhancing the BCR accumulation on soft substrates (Figure 9—figure supplement-2 A-I).

[Editors' note: further revisions were requested prior to acceptance, as described below.]

*The manuscript has been improved but there are some remaining issues that need to be addressed before acceptance, as outlined below:*

*For the new data and Discussion provided in the revised manuscript, there two suggestions.*

*1) The authors should point out that while EDTA can interfere with integrin-mediated cell adherence, as a calcium chelator, EDTA may have a much broader impact on B cells, including BCR-induced calcium influx.*

We fully agree with the reviewers that although EDTA can interfere with the function of integrin, EDTA is also known as an extremely potent calcium chelator with awesome divalent binding capability. Thus, EDTA has been demonstrated to have a much broader impact on the activation and function of B cells, including BCR-antigen binding mediated calcium influx. Due to all these promiscuity effects from EDTA to the activation and function of B cells, we also discussed in the revised manuscript that the results from the EDTA-treated cells would not allow us to draw any solid conclusion in terms of the contribution of integrin to the substrate stiffness discrimination capability of B cells during the initiation. Thus, we also examined the responses of the splenic primary B cells from CD11a (ITGAL) knockout (KO) mice, which lack the lymphocyte function-associated antigen 1 (LFA-1). These results from LFA-1 KO B cells shall be more convincing for the establishment of the conclusion that B cell discrimination against substrate stiffness is dependent on adhesion molecules (Figure 8).

*2) The new data that show that B cells discriminate stiff and soft antigen presenting surface by increasing BCR clustering at the stiff surface, which is dependent on the proximal signaling of the BCR. In contrast, B cells from lupus patients fail to increase BCR clustering on the stiff surface while enhancing BCR clustering on the soft surface. This result should be included in the discussion of signaling-dependent and independent BCR clustering and its implication in B cell activity in lupus patients.*

This is also an excellent point. We thank the reviewers for this constructive advice. We have now improved the Discussion section by including those specific results suggested by reviewers. In brief, we mentioned that both signaling-independent and signaling-dependent mechanisms account for the accumulation of BCRs into the BCR microclusters. These two steps are not mutually exclusive, but are instead closely related in a sequential and synergistic way to maximize efficient BCR clustering. Specifically, signaling-independent BCR clustering initiates the earliest signaling upon antigen and BCR recognition, which further enhances BCR clustering in a signaling-dependent manner. Our data in this report show that the initial signaling-independent accumulation of BCRs into the BCR microclusters is not sensitive to the stiffness features of the substrates whereas the subsequent signaling-dependent accumulation of BCRs into the BCR microclusters is. We examined whether the loss of substrate stiffness discrimination ability is due to the changes in the BCR MFI (as a parameter indicating BCR microclustering) on a stiff or soft substrate surface. We achieved this by comparing the BCR MFI of the inhibitor pre-treated versus DMSO-control pre-treated primary B cells on the surface of the substrates with the same Young's modulus. The results readily demonstrated that inhibition of proximal signaling molecules, Lyn, Syk or PLCγ2 primarily blocked the synaptic accumulation of BCRs on stiff substrates, while the changes of BCR MFI on soft substrates were very mild in the comparison of inhibitor pre-treated versus DMSO-control pre-treated primary B cells. Furthermore, we examined whether the alterations in the ability of RA patient B cells to discriminate substrate stiffness of antigen-presenting surface are due to the changes in BCR MFI on a stiff or soft substrate surface. The results showed that B cells from healthy controls preferentially enhanced the synaptic accumulation of BCR microclusters on stiff substrates, while RA patient B cells exhibited different preference of enhancing the BCR accumulation on soft substrates. Since B cells from RA patients are known to exhibit hyper BCR signaling (Nakken et al., 2011, Autoimmun Rev; Oligino and Dalrymple, 2003, Arthritis Research and Therapy; Szodoray et al., 2006, Rheumatology,). We speculate that the enhanced activation of RA patient B cells on soft substrates results from the signaling-dependent accumulation of BCRs into the BCR microclusters as discussed above in this report.

*The authors are urged to have the paper read and edited carefully for correct English language usage. We anticipate that a manuscript including these specific changes and with some overall improvement in English editing will be acceptable for publication without a need for re-review. So please do your best in order to minimize delays associated with extensive copyediting.*

We agree and we have used the help from a professional English language service provider to thoroughly edit this version of the manuscript.